# Underreporting of SARS-CoV-2 infections during the first wave of the 2020 COVID-19 epidemic in Finland—Bayesian inference based on a series of serological surveys

**Tuomo A. Nieminen**[1,2]\*, **Kari Auranen**[3], **Sangita Kulathinal**[2], **Tommi Härkänen**[1,2], **Merit Melin**[1], **Arto A. Palmu**[1], **Jukka Jokinen**[1,2]

**1** Finnish Institute for Health and Welfare, Helsinki, Finland, **2** University of Helsinki, Helsinki, Finland, **3** University of Turku, Turku, Finland

\* tuomo.nieminen@thl.fi

## Abstract

In Finland, the first wave of the COVID-19 epidemic caused by the severe acute respiratory syndrome coronavirus 2 (SARS-CoV-2) took place from March to June 2020, with the majority of COVID-19 cases diagnosed in the Helsinki-Uusimaa region. The magnitude and trend in the incidence of COVID-19 is one way to monitor the course of the epidemic. The diagnosed COVID-19 cases are a subset of the infections and therefore the COVID-19 incidence underestimates the SARS-CoV-2 incidence. The likelihood that an individual with SARS-CoV-2 infection is diagnosed with COVID-19 depends on the clinical manifestation as well as the infection testing policy and capacity. These factors may fluctuate over time and the underreporting of infections changes accordingly. Quantifying the extent of underreporting allows the assessment of the true incidence of infection. To obtain information on the incidence of SARS-CoV-2 infection in Finland, a series of serological surveys was initiated in April 2020. We develop a Bayesian inference approach and apply it to data from the serological surveys, registered COVID-19 cases, and external data on antibody development, to estimate the time-dependent underreporting of SARS-Cov-2 infections during the first wave of the COVID-19 epidemic in Finland. During the entire first wave, there were 1 to 5 (95% probability) SARS-CoV-2 infections for every COVID-19 case. The underreporting was highest before April when there were 4 to 17 (95% probability) infections for every COVID-19 case. It is likely that between 0.5%–1.0% (50% probability) and no more than 1.5% (95% probability) of the adult population in the Helsinki-Uusimaa region were infected with SARS-CoV-2 by the beginning of July 2020.

## Introduction

When a novel virus initiates an epidemic, an important question is how fast the virus spreads in the population. If the virus causes specific clinical disease, the rate of epidemic growth can be monitored by the incidence of diagnosed disease cases. However, mild or asymptomatic

**Data Availability Statement:** Aggregated data which supports the findings of the study are included within the paper. Data sufficient to

approximately reproduce the main results are also available online in machine readable format (csv) here: github.com/TuomoNieminen/ covid19underreporting. Individual level data are not available due to data privacy by Finnish law. More information may be requested from the Finnish Institute for Health and Welfare by contacting kirjaamo@thl.fi.

**Funding:** This study was funded internally by the Finnish Institute for Health and Welfare with state budget item for SARS-CoV-2 studies. The authors were employees of the Finnish Institute for Health and Welfare, but the funders had no role in study design, data collection and analysis, decision to publish, or preparation of the manuscript.

**Competing interests:** We report no conflict of interests related to the current work. The Finnish Institute for Health and Welfare (THL) conducts Public-Private Partnership with vaccine manufacturers and has received research funding from Sanofi Inc., Pfizer Inc., and GlaxoSmithKline Biologicals SA for studies not related to COVID-19. Nieminen, Melin, Palmu and Jokinen have been investigators in these studies but they have received no personal remuneration. These projects are not directly related to the current work and do not alter our adherence to PLOS ONE policies on sharing data and materials.

infections may be difficult or impossible to observe directly, and therefore the true incidence of infection can not be learned solely based on the diagnosed cases. Infection usually leaves a mark in the form of antibodies, i.e. immunoglobulin proteins developed by the immune system and capable of identifying and neutralising the virus. Consequently, the true incidence of infection can be learned through serological surveys, i.e. studies of the prevalence of individuals with antibodies (seroprevalence). Comparing the seroprevalence to the cumulative incidence of diagnosed cases allows one to learn about the underreporting of infections, which consequently allows monitoring the true spread of the virus.

There are challenges in estimating the level of underreporting. The rate of infections and diagnostic practises may quickly change, and there may be different delays from infection to disease onset and to developing antibodies. In this case study, we propose a Bayesian approach for estimating the time-dependent underreporting of infections during the beginning of an epidemic and we apply our method to data from the 2020 COVID-19 epidemic in Finland. In our analysis we integrate three data sources: series of serological surveys, registered disease cases, and external data on antibody development.

In Finland, the first wave of the COVID-19 epidemic caused by the severe acute respiratory syndrome coronavirus 2 (SARS-CoV-2) occurred from March through June 2020. In early March, tens of weekly COVID-19 cases were diagnosed in the Helsinki-Uusimaa region (HUS area) with a population of 1.7 million, marking the beginning of the epidemic in the region, while relatively few cases were diagnosed in other parts of the country. Fig 1 shows the numbers of new COVID-19 cases by week and municipality in the HUS area. Already by mid March, hundreds of weekly cases were diagnosed. The rate of new cases started to decline in early April, most likely because of a partial lockdown in the country. By mid June, the rate of weekly cases, both in the HUS area and the country as a whole, reduced to the tens of cases, and the first wave of the COVID-19 epidemic ended by the end of June. A total of 7286 COVID-19 cases were diagnosed during the first epidemic wave, of which 5347 cases were diagnosed in individuals residing in the HUS area.

The clinical manifestations of SARS-CoV-2 infection range from asymptomatic to severe and potentially fatal disease. To be diagnosed as a COVID-19 case, a SARS-CoV-2 infection needs to be laboratory confirmed or, alternatively, a clinical diagnosis of COVID-19 made by a medical doctor. The likelihood of a SARS-CoV-2 infection being detected thus depends on the clinical manifestation as well as the infection testing policy and capacity at the time of infection.

It is likely that a relatively large proportion of infections went undetected during the first wave of the epidemic in Finland. No widespread testing of asymptomatic individuals was in place, making it probable that at least almost all asymptomatic infections were missed. Many symptomatic infections were likely missed as well due to the care instructions and testing policy in place. In Finland, the underreporting was probably most prominent among the young and healthy in the beginning of the first epidemic wave, when the official care instructions for healthy individuals with symptoms compatible with COVID-19 were to stay at home with no contact to health care [1]. These instructions were affected by the limited infection testing capacity. During the epidemic peak, the daily number of infection tests in the HUS area was still increasing through rapid capacity building. The daily testing capacity increased from approximately 300 during March to 1000 during April to 1500 tests from May onward [2].

Based on a population-based seroepidemiological study in Spain in April-May 2020, Pollán et al. found that approximately one third of SARS-CoV-2 infections were asymptomatic and that a substantial proportion of symptomatic infections also went undetected [3]. Stringhini et al. analysed the prevalence of immunoglobulin G (IgG) antibodies in Geneva during spring

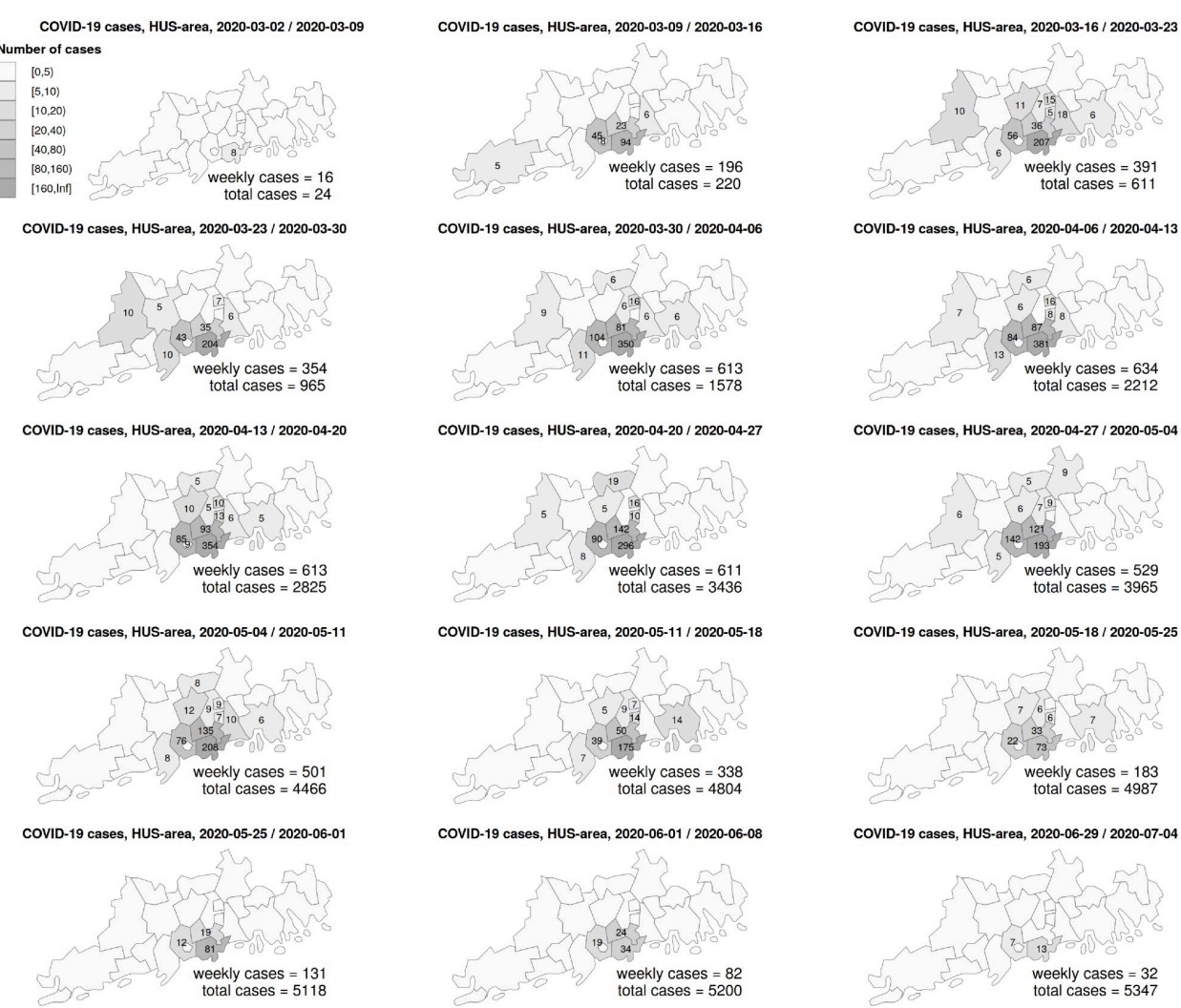

**Fig 1. Numbers of COVID-19 cases by week and municipality in the Helsinki-Uusimaa region during the first wave of the 2020 COVID-19 outbreak.** In each map, the number of cases in each municipality is shown if it is 5 or more. Detailed data for three weeks in June (2020-06-08 / 2020-06-15, 2020-06-15 / 2020-06-22 and 2020-06-22 / 2020-06-29), with total 147 cases, are not shown in the figure.

2020 and estimated that there were 11 SARS-Cov-2 infections for every detected COVID-19 case [4]. Erikstrup et al. analysed blood donation data in April-May 2020 in Denmark and estimated that the ratio of the expected number of seropositives to the number of COVID-19 cases was between 7–20 [5]. To obtain information on the incidence of SARS-CoV-2 infection in Finland, a series of serological surveys (serosurveys) was initiated in April 2020.

While there may be significant delays from SARS-CoV-2 infection until developing detectable antibodies, i.e. until seroconversion, symptoms and diagnosis of COVID-19 usually occur with less delay. This means that the two sources of observations are not directly comparable at any given time. One solution to this problem is to compare the SARS-CoV-2 seroprevalence to the cumulative incidence of COVID-19 from 2–3 weeks ago, thus accounting for the average delay in developing antibodies after COVID-19 symptoms. This approach can provide an estimate of underreporting but it does not take into account the uncertainty and individual-level variation in the time lag from COVID-19 symptoms to seroconversion.

To better address the delays in antibody responses, in this paper, we utilise previously published data about the time lag from COVID-19 symptom onset to seroconversion [6]. We estimate the distribution of the time lag and project the SARS-CoV-2 seroprevalence based on the COVID-19 incidence. We then estimate the SARS-CoV-2 seroprevalence based on the observations from the series of serosurveys. Finally, we compare the seroprevalence projections with the estimated seroprevalences over time and learn the time-evolving underreporting of SARS-CoV-2 infections based on the ratio of the two measures of seroprevalence.

We utilise Bayesian inference and data from the HUS area to carry out the analysis. The novelty of our methodology is in accounting for the uncertainty in the time lag from disease symptoms to seroconversion when estimating the time-evolving underreporting of infections. Our analysis shows how the underreporting of SARS-CoV-2 infections evolved over time during the first epidemic wave in Finland.

## Data sources

### Study population

The target population in this study include individuals aged 18–69 years and living in the HUS area with native language Finnish, Swedish, English or Russian. We utilised the Finnish Population Information System (PIS) to retrieve the native languages of all COVID-19 cases and the serological survey participants. We also retrieved the age distribution of the study population from the same system. The PIS includes the Finnish personal identity code, birth date, native language and municipality of residence for all Finnish residents [7]. We present some data for the whole HUS area population, but our main analysis is based on data from the study population.

### COVID-19 cases

The Finnish National Infectious Diseases Register (FNIDR) collects individual-level data on patients infected with SARS-CoV-2 [8]. These data consist of COVID-19 cases notified as either a positive SARS-CoV-2 finding from a microbiological laboratory or a clinical diagnosis by a medical doctor. Approximately 95% of the COVID-19 cases during the first epidemic wave in Finland were based on a positive SARS-CoV-2 finding from a polymerase chain reaction (PCR) test. The data was extracted for analysis on 31st November 2021.

The sample date of each positive PCR test and/or a doctor's diagnosis is recorded in the FNIDR along with information regarding the patient, including the Finnish personal identity code. Records related to the same patient during a 12-month period are combined as a single COVID-19 case. In our analysis, the COVID-19 diagnosis date is taken to be the first positive PCR sample date or the first doctor's diagnosis date, whichever occurred first. According to expert evaluation during early 2020, the delay from symptom onset to COVID-19 diagnosis was deemed to be on average 3.5 days in the Helsinki-Uusimaa region.

### Serological surveys

In April 2020, the Finnish Institute for Health and Welfare (THL) initiated a series of serological surveys (serosurveys) to obtain information on how large a proportion of the population had developed antibodies to SARS-CoV-2 in different regions in Finland over time [9]. Each survey targeted most of the largest municipalities in Finland and individuals aged 18–69. In each survey round, individuals were randomly sampled from PIS and invited to participate. Successive surveys were conducted weekly or biweekly.

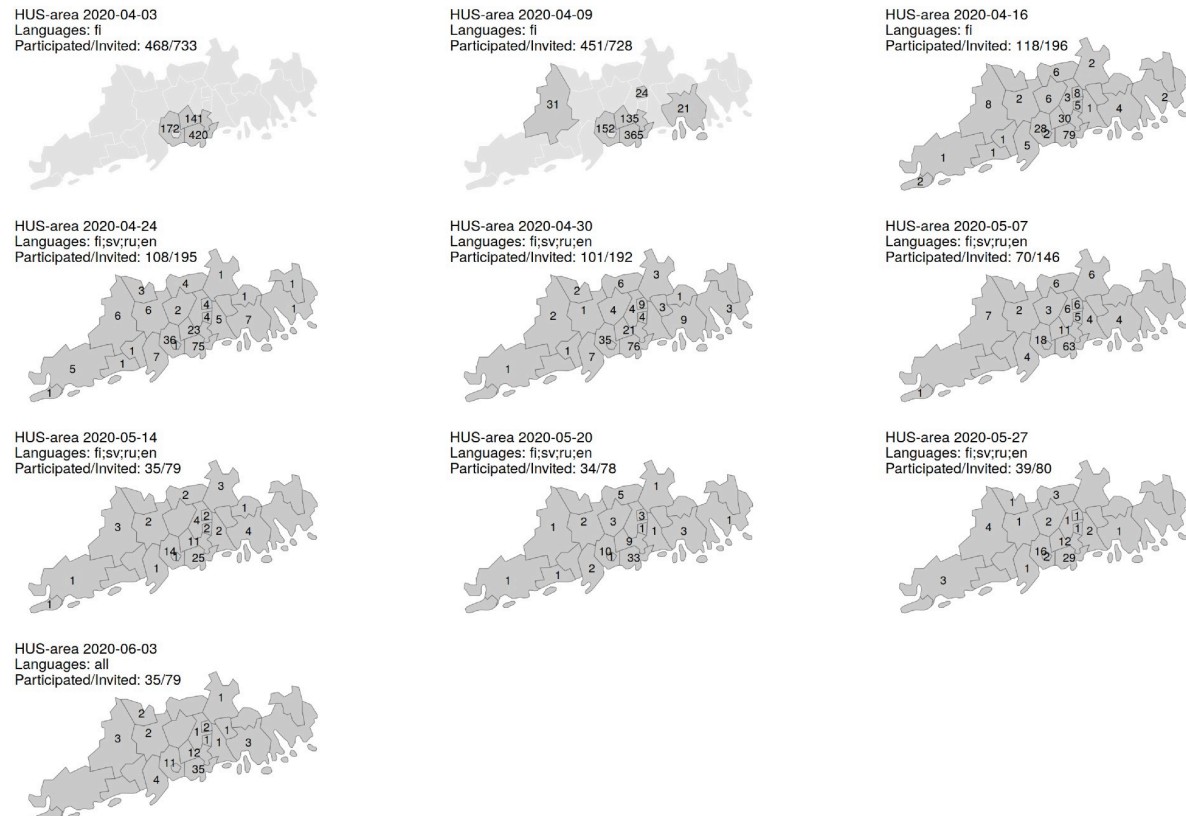

**Fig 2. Population sampling in the Helsinki-Uusimaa region during the first 10 weeks of the serological surveys.** Population sampling was carried out weekly or biweekly and each map corresponds to a single sampling week. The first day of the week and the targeted native languages are listed for each sampling week (fi = Finnish, sv = Swedish, ru = Russian, en = English). The number of invited individuals by municipality are shown on each map.

Fig 2 shows the recruitment to and participation in the surveys in the HUS area during the spring 2020. Due to practical reasons, only Finnish speaking individuals were recruited during the first two weeks, after which the study expanded to cover individuals with native language Swedish, English or Russian. The questionnaire was translated to each language. Other language groups were included in June 2020. The recruitment targeted only few of the largest municipalities during the first two weeks and then expanded to cover all municipalities in the HUS area. The sample size in the HUS area decreased after the second week and the participation rate declined from 64% to 50% during spring 2020.

The age distribution of the study population and the survey participants during the first epidemic wave are shown in S1 Fig. The median and the 25% quantile of the age of the survey participants were slightly higher than in the study population, indicating that the participation rate was higher in older age groups. Otherwise the age distribution of the participants was similar to the study population.

Participation in the survey included giving a blood sample. The first and last blood samples during the first epidemic wave were taken on 9th April 2020 and 3rd July 2020, respectively.

## Laboratory methods

Blood samples from the serosurvey participants were analysed using a two stage procedure: (1) a screening test, and (2) a microneutralisation test (MNT) following a positive result at stage

(1). The screening test was a bead-based fluorescence immunoassay that measures IgG antibodies to the SARS-CoV-2 nucleoprotein [10]. The MNT is a cytopatic effect-based test, which measures the capacity of neutralising antibodies to prevent an infectious virus from causing damage in cell culture. SARS-CoV-2 strains circulating in Finland in early 2020 were used in the MNT assay; CoV-19/Finland/1/2020 (GISAID accession ID EPI_ISL_407079) and hCoV-19/Finland/FIN-25/2020 (EPI_ISL_412971). MNT was used as the second test as it is highly specific to SARS-CoV-2 [10, 11]. Obtaining positive results from the two tests, the screening test and the MNT combined, was considered a confirmed presence of antibodies due to a past or ongoing SARS-CoV-2 infection (seroconversion). In the following, the combined test is referred to as the confirmation test.

In order to maximise accuracy, the confirmation test was calibrated utilising data unrelated to the surveys [10]. The ground truth for a past or ongoing SARS-CoV-2 infection was based on a positive PCR test close to 30 days prior to the antibody tests. The ground truth of no SARS-Cov-2 was based on blood samples from 2019. Based on calibration, a sample was considered positive for the screening test if the mean fluorescent intensity (MFI) value of the test was above 500. In the MNT, neutralising antibodies were detected from 2-fold serially diluted serum samples starting from dilution 1:4. Based on calibration, a titer of $\geq 4$ was considered positive. S2 Fig describes the optimised test performance on the calibration data for both the screening and confirmation tests. The screening test was 100% sensitive, after which the MNT was both 100% specific and 100% sensitive. Therefore the optimised performance of the confirmation test was 100% sensitive and 100% specific. The sensitivity and specificity of the screening test alone were 100% and 97.59%, respectively.

## Development and detection of antibodies

For the screening test, we say that an individual is *seropositive* if the test gives a positive result. If the seropositivity is due to a SARS-CoV-2 infection, we say that the individual is *seroconverted*. An individual may be seropositive but not seroconverted, because the screening test may produce a false positive result due to cross-reactive IgG antibodies induced by other human coronaviruses. Neutralising antibodies measured by the MNT are always due to SARS-CoV-2 and therefore and individual with a positive confirmation test is always both seropositive and seroconverted.

The time from infection to seroconversion is subject to individual-level variation. If time from infection is short, the antibody concentration may not have reached the test detection threshold. If time from infection is long, the antibodies may wane below the detection threshold. The sensitivity of antibody detection (e.g. the confirmation test) is therefore likely to be lower than 100% in both of these cases. When modelling the time-dependent seroconversion, we take into account the slow development of antibodies after infection. However, we omit waning immunity due to the relatively short study period.

For symptomatic individuals, the symptoms usually develop sooner than detectable antibodies. Tan et al. present results where symptomatic SARS-CoV-2 infected patients were followed for 6 weeks starting from symptom onset and reported the IgG positive proportions of patients for each week [6]. The antibody test utilised in their analysis was similar to the screening test of the current study. The data are reproduced in Table 1. A total 312 tests were performed on 65 patients, with 3–7 days between consecutive tests. At day 7 since symptom onset, only 3.4% of the patients tested positive for IgG antibodies. At day 14, 50% tested positive and when 28–49 days had passed, between 74% and 87% tested positive. Tan et al. report that of the 67 patients included in their study, 29 were classified with severe pneumonia [6]. The median age of the patients was 49 years and twenty-five patients had underlying diseases.

**Table 1. Percentage of seroconverted COVID-19 patients by time since symptom onset.**

| Day | Patients | IgG positive | % |
|---|---|---|---|
| 7 | 58 | 2 | 3.4 |
| 10 | 62 | 12 | 19.4 |
| 14 | 61 | 31 | 50.8 |
| 21 | 54 | 32 | 59.3 |
| 28 | 35 | 26 | 74.3 |
| 35 | 22 | 17 | 77.3 |
| 42 | 15 | 13 | 86.7 |
| 49 | 5 | 4 | 80.0 |

A total 312 tests were performed on 65 patients. Day is the number of days passed since COVID-19 symptom onset, Patients are the number of patients tested and IgG positive are the number of patients who tested positive for SARS-CoV-2 IgG antibodies. Data from Tan et al. [6].

## Statistical models and methods

Let $T = [0, D]$, $D = 86$, denote the study period, i.e. the time period starting on 9th April 2020 (the date of the first blood sample taken from the serosurvey participants), until 3rd July 2020 (the date of the last blood sample taken during the first epidemic wave). Let $\tau_i$ denote the day of SARS-CoV-2 infection in individual $i$, $i = 1, \ldots, N$. Here $N = 1000821$ is the size of the study population. The infections we consider may have occurred before the study period but not after (i.e. $\tau_i$ may be negative and $\tau_i < D$).

After the infection, on day $s_i$, the individual may develop symptoms of COVID-19. Then, $C$ days after the symptom onset, on day $r_i = s_i + C$, the individual may be diagnosed with COVID-19. In this case, information about the diagnosis and the individual is recorded in the FNIDR as a COVID-19 case. We assume that the delay $C$ from symptom onset to diagnosis is 3.5 days and is the same for all individuals. The cumulative number of COVID-19 cases by day $t$ is $R(t)$, where $R(t) = \sum_i^N 1(r_i \leq t)$.

An individual $i$ has *seroconverted* by day $t$ if $t > a_i > \tau_i$, where $a_i$ is the day after which the SARS-CoV-2 antibodies in the individual are detectable. We define $A_i(t) = 1(a_i < t)$ as an indicator function taking value 1 for individual $i$ if seroconversion has occurred by day $t$ and 0 otherwise. For individuals with diagnosed COVID-19 we assume that seroconversion occurs after the symptom onset day (i.e. $a_i > s_i$). In those cases, we use $U_i$ to denote the number of days from symptom onset to seroconversion. Fig 3 summarises the notation and describes the timeline from SARS-CoV-2 infection to seroconversion.

Regardless of the infection status, an individual from the study population may be randomly selected to participate in one of the serosurveys. Let $y_{i,t}^{(z)} \in \{0, 1\}$ denote the binary test result (i.e. seropositivity) for individual $i$ who was randomly selected into the survey and gave a sample for antibody testing on day $t \in T$, where $z \in \{\text{Screen, Confirmation}\}$ denotes the test used to derive the result. We denote the specificity of test $z$ as $\delta^{(z)} = \Pr(y_{i,t}^{(z)} = 0 \mid \tau_i > t)$. If the test $z$ is not fully specific, i.e. $\delta^{(z)} < 1$, then the result may be positive ($y_{i,t}^{(z)} = 1$) without a SARS-COV-2 infection.

Fig 4 displays how SARS-CoV-2 infections may have been observed as COVID-19 cases or positive antibody test results. To compare estimates of seroprevalence based on the two types of observation (serosurveys and COVID-19 cases), we quantify the distribution of the time lag from COVID-19 symptom onset to seroconversion. We then project the time-dependent seroprevalence based on the diagnosed COVID-19 cases, which allows for comparison to the seroprevalence estimated from the serosurveys.

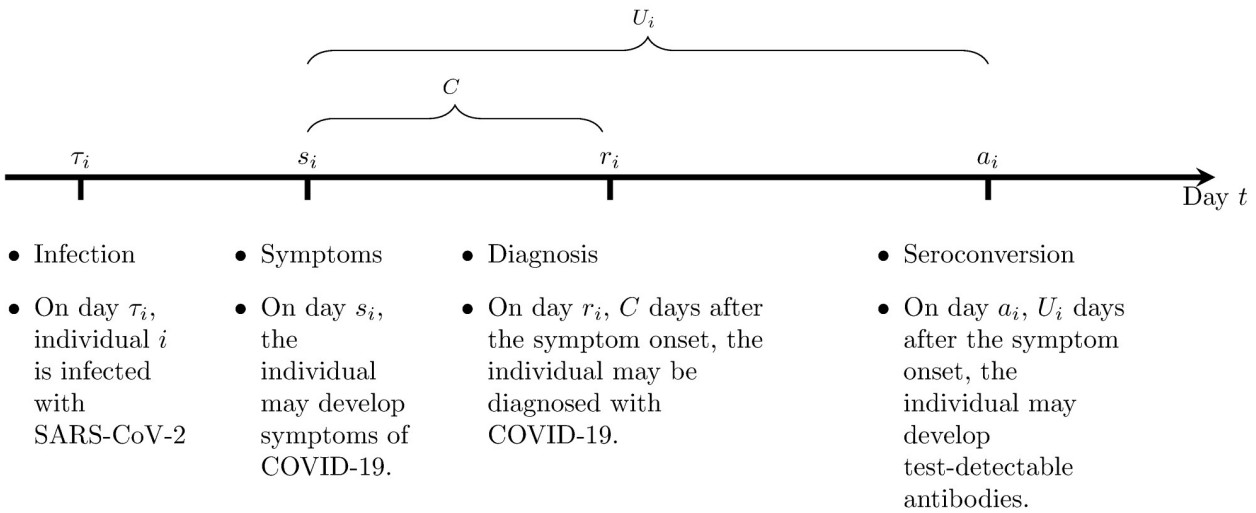

**Fig 3. Timeline from a SARS-CoV-2 infection to seroconversion.**

## Estimation target

Under two independent models, the quantity of interest is seroprevalence $\pi(t)$, i.e. the proportion of the population that has seroconverted by time $t$, where $\pi(t) = \Pr(A_i(t) = 1) = \mathbb{E}(A_i(t))$, for $i = 1, \ldots, N$. We estimate $\pi(t)$ using (i) observations from the serosurveys and (ii) the incidence of COVID-19 cases. We denote $\pi^{(0)}(t)$ to indicate the seroprevalence when based on the serosurveys and $\pi^{(1)}(t)$ when based on COVID-19 cases. Our interest is in estimating the ratio of these two seroprevalence parameters on each day $t \in T$ during the study period:

$$\Delta(t) = \frac{\pi^{(0)}(t)}{\pi^{(1)}(t)}. \tag{1}$$

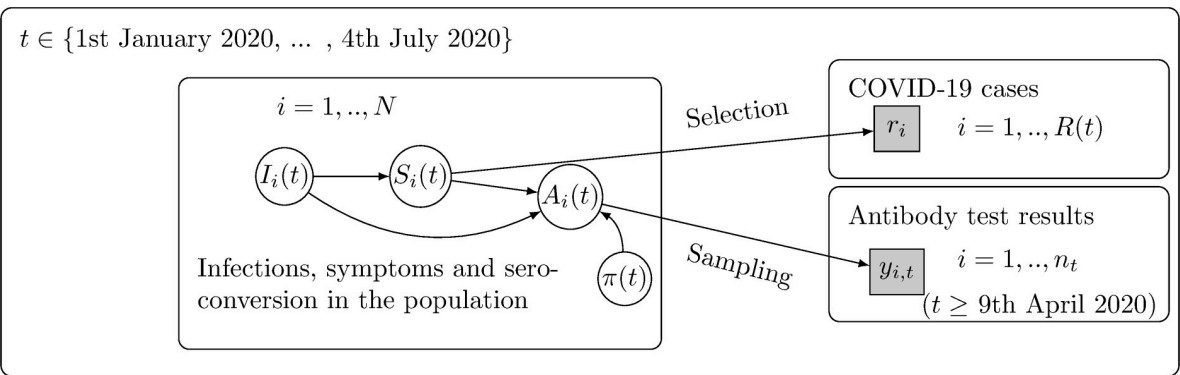

**Fig 4. Observations related to SARS-CoV-2 infections.** SARS-CoV-2 infections are observed as diagnosed COVID-19 disease cases or by antibody testing in the participants of the serological surveys. Here, $I_i(t)$ indicates SARS-CoV-2 infection in individual $i$ by day $t$; $S_i(t)$ indicates symptom onset in individual $i$ by day $t$; $r_i$ is the diagnosis day of a COVID-19 case, with a total $R(t)$ cases by day $t$; $A_i(t)$ indicates seroconversion in individual $i$ by day $t$, possibly observed as a positive antibody test $y_{i,t} = 1$ among $n_t$ blood samples taken on day $t$; $\pi(t)$ indicates the proportion of seroconverted individuals in the population of size $N$ (seroprevalence).

We estimate $\pi^{(0)}(t)$ and the corresponding $\Delta(t)$ separately for data from the two antibody tests but consider the analysis based on the confirmation test as the main result. In section Models we describe an *Estimation model* used to estimate $\pi^{(0)}(t)$ and a *Projection model* used to estimate $\pi^{(1)}(t)$. We expect that $\pi^{(0)}(t)$ gives a reasonably unbiased estimate of the true seroprevalence $\pi(t)$ but expect that the projection $\pi^{(1)}(t)$ gives an underestimate of the true $\pi(t)$. We therefore expect that $\Delta(t) > 1$ and interpret $\Delta(t)$ as an underreporting ratio, i.e. quantifying the extent of underreporting of SARS-CoV-2 infections up until time $t$.

## Models

In this section, we specify the Estimation and Projection model of the seroprevalence. We then describe the estimation of seroprevalence and underreporting under both models. We utilise a Bayesian framework for statistical inference and numerical methods to derive the posterior distributions of all unknown quantities.

**Estimation model.** This model relates to the lower part of Fig 4 (Sampling). The Estimation model is used to estimate the time-dependent seroprevalence based on antibody test results in the serosurvey participants. Due to the small numbers of daily blood samples in the serosurveys, we split the study period $T$ into 13 non-overlapping seven day periods (weeks), $W = [0, 7), [7, 14), ....[84, 86]$. We assume that the seroprevalence is piecewise constant by week and let $\pi_w^{(0)}$ denote the seroprevalence during week $w \in W$.

We describe the prior uncertainty in the weekly seroprevalence as follows. For the first week, the logit of the seroprevalence $g(\pi_1^{(0)})$ is assumed to be normally distributed with expectation $\mu_1$ and variance $\sigma_1^2$. Note that the normal distribution is the maximum entropy distribution for $g(\pi_1^{(0)})$ under the constraints that its expectation is $\mu_1$ and variance is $\sigma_1^2$. The logit of the prevalence in any later week is assumed to depend on the prevalence during the previous week with a non-decreasing trend. A shared variance parameter $\sigma^2$ (which is different from $\sigma_1$) controls the strength of the dependency on the previous weeks, with $\sigma$ given a gamma prior with parameters $\alpha$ and $\beta$. The structure of the prior model thus is:

$$\sigma \sim \text{Gamma}(\alpha, \beta),$$

$$g(\pi_1^{(0)}) \sim N(\mu_1, \sigma_1^2),$$

$$g(\pi_w^{(0)}) \sim N(g(\pi_{w-1}^{(0)}) + \text{trend}_w, \sigma^2) \quad \text{for } w \geq 2, \text{ where} \tag{2}$$

$$\text{trend}_w = \begin{cases} 0, \text{ when } w = 2 \\ \max\{0, g(\pi_{w-1}^{(0)}) - g(\pi_{w-2}^{(0)})\}, \text{ when } w > 2, \end{cases}$$

where $g(\pi) = \log(\pi/(1 - \pi))$ is the logit function. This defines a prior distribution of the parameter vector $g(\pi^{(0)}) = (g(\pi_1^{(0)}), .., g(\pi_{13}^{(0)}))$. We denote the prior distribution of $g(\pi^{(0)})$ as $p(g(\pi^{(0)}); \Phi)$, where the vector $\Phi = (\alpha, \beta, \mu_1, \sigma_1)$ collects the hyperparameters. The seroprevalence for week $w$ is $\pi_w^{(0)} = g^{-1}(g(\pi_w^{(0)}))$, where $g^{-1}(x) = 1/(1 + \exp(-x))$ is the inverse-logit function.

The observations $y_{i,w}^{(z)} \in \{0, 1\}$ arise when $n_w$ randomly selected individuals from the population give a blood sample during week $w$ and a result is derived via antibody test $z$. The probability that the test result is positive for individual $i$ is

$$\begin{aligned} \Pr(y_{i,w}^{(z)} = 1) \quad &= f(\pi_w^{(0)}, \delta^{(z)}) \\ &= \pi_w^{(0)} + (1 - \pi_w^{(0)}) \cdot (1 - \delta^{(z)}), \end{aligned} \tag{3}$$

where $1 - \delta^{(z)}$ is the probability that an individual without SARS-CoV-2 infection gives a (false) positive test result.

Let $y_w^{(z)} = \sum_{i=1}^{n_w} y_{i,w}^{(z)}$ denote the number of positive samples during week $w$. We assume that, conditionally on the weekly seroprevalence, the observations $y_{i,w}^{(z)}$ are independent and identically distributed. The conditional probability model of the total count $y_w^{(z)}$, where $w \in W$, then is

$$y_w^{(z)} \mid g(\pi_w^{(0)}); \delta^{(z)} \sim \mathrm{Binom}(n_w, f(\pi_w^{(0)}, \delta^{(z)})). \tag{4}$$

Based on the vector of observations $\mathbf{y}^{(z)} = (y_1^{(z)}, ..., y_{13}^{(z)})$, the likelihood function of the logit seroprevalence $g(\pi^{(0)})$ is

$$p(\mathbf{y}^{(z)} \mid g(\pi^{(0)}); \delta^{(z)}) = \prod_{w \in W} \mathrm{Binom}(y_w^{(z)} | n_w, f(\pi_w^{(0)}, \delta^{(z)})). \tag{5}$$

The posterior distribution of $g(\pi^{(0)})$ is proportional to the product of the prior (2) and the likelihood (5):

$$p(g(\pi^{(0)}) \mid \mathbf{y}^{(z)}; \Phi, \delta^{(z)}) \propto p(g(\pi^{(0)}); \Phi) p(\mathbf{y}^{(z)} \mid g(\pi^{(0)}), \delta^{(z)}). \tag{6}$$

The estimation model is described graphically in Fig 5. We defined an informative prior distribution for the Estimation model seroprevalence. The chosen hyperparameter values $\mu_1 = logit(0.05)$ and $\sigma_1 = 2$ correspond to an approximate prior expectation 0.13 for the seroprevalence at the start of the study but with large variance. The chosen hyperparameter values $\alpha = 2$ and $\beta = 40$ correspond to expected value of approximately 0.5 for the standard deviation between weekly seroprevalences on the probability scale. S3 Fig shows the prior distribution for $\pi^{(0)}$. In the prior distribution, each weekly seroprevalence $\pi^{(0)}(t)$ has a large variance. The prior mean and variance both increase as $t$ increases.

**Projection model.** This model relates to the upper part of Fig 4 (Selection). The model is learned from previously published data on antibody development after COVID-19 symptoms.

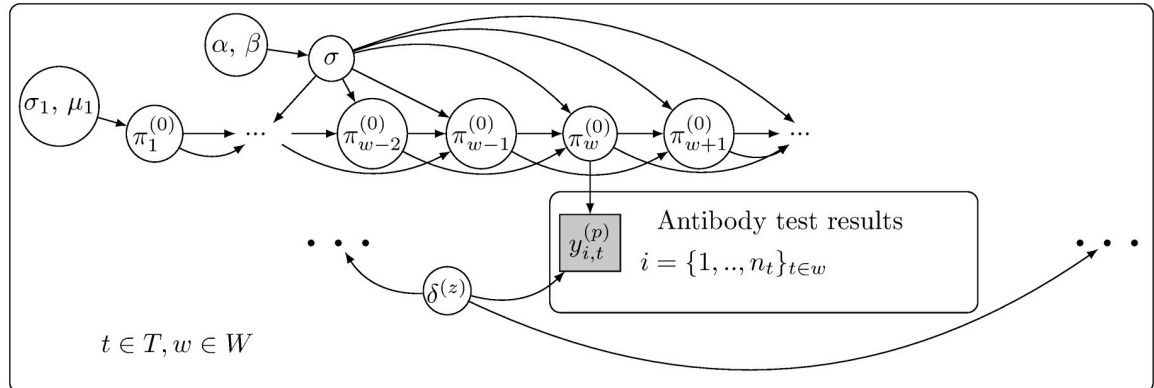

**Fig 5. The model for seroprevalence $\pi^{(0)}(t)$ (estimation model).** The study period $T$ is split into weeks $W$. On day $t$ during the study period, where $t$ belongs to week $w$, the antibody test result $y_{i,t}^{(z)}$ for individual $i$, $i = 1, \ldots, n_{t \in w}$, depends on the seroprevalence $\pi_w^{(0)}$ during week $w$ and the specificity $\delta^{(z)}$ of the antibody test $z$, where $z \in \{\text{Screen, Confirmation}\}$. In the prior distribution, the seroprevalence during the first week $\pi_1^{(0)}$ is distributed according to parameters $\mu_1$ and $\sigma_1^2$, and the seroprevalence during week $w$ depends on the two previous weeks. The strength of the dependency is controlled by $\sigma$, with a prior distribution depending on parameters $\alpha$ and $\beta$.

We first describe the model and then show how it is utilised to project the time-dependent seroprevalence based on COVID-19 cases in the FNIDR.

For individual $j$, the number of days from COVID-19 symptom onset to seroconversion is described by the random variable $U_j$. We assume that each $U_j$ has a lognormal distribution with parameters $\mu_U$ and $\sigma_U^2$. The probability that patient $j$ has secoconverted by day $u$ since symptom onset is $\Pr(U_j \le u) = F_U(u; \theta)$, where $\theta = (\mu_U, \sigma_U^2)$.

To estimate the parameters $\theta$, we utilise data based on patients who had SARS-CoV-2 antibodies tested on multiple days after COVID-19 symptoms [6]. The data are shown in Table 1. We denote the test result by $y_j^q \in \{0, 1\}$ for individuals $j = 1, \ldots, n^q$, where $n^q$ is the number of individuals tested $q$ days after symptom onset, and $q \in Q^{Tan} = \{7, 10, 14, \ldots, 42, 49\}$. If the test result is positive (i.e. $y_j^q = 1$), the patient is seroconverted and the seroconversion must have occurred before day $q$. The probability model for the individual observation is

$$y_j^q \mid \theta \sim \mathrm{Bern}(F_U(q; \theta)). \tag{7}$$

We assume that the test results are independent given day $q$ and the parameters $\theta$. Based on the observations $\mathbf{y}^{Tan} = \{y_j^q, j = 1, \ldots, n^q, q \in Q^{Tan}\}$, the likelihood function of the parameters $\theta$ is

$$p(\mathbf{y}^{Tan} \mid \theta) = \prod_{q \in Q^{Tan}} \prod_{j=1}^{n^q} \mathrm{Bern}(y_j^q | F_U(q; \theta)). \tag{8}$$

We assume an uninformative prior distribution:

$$p(\theta) = p(\mu_U, \sigma_U^2) \propto 1/\sigma_U^2. \tag{9}$$

The posterior distribution is proportional to the product of the prior (9) and the likelihood (8):

$$p(\theta \mid \mathbf{y}^{Tan}) \propto p(\theta) p(\mathbf{y}^{Tan} \mid \theta). \tag{10}$$

The posterior predictive distribution of $F_U$ is

$$\hat{F}_U(u) = p(y_j^u \mid \mathbf{y}^{Tan}) = \int F_U(u; \theta) p(\theta \mid \mathbf{y}^{Tan}) d\theta. \tag{11}$$

We utilise the posterior predictive distribution $\hat{F}_U$ to project seroprevalence based on the FNIDR COVID-19 cases. For each day $t \in T$ during the study period, we first predict the probability of seroconversion in each case $i$, for whom $q_i$ days have passed since symptom onset. We assume that the symptom onset day was $C = 3.5$ days before the diagnosis day $r_i$, and so $q_i = t - (r_i - C)$. The probabilities of seroconversion, each given by $\hat{F}_U(q_i)$, are then combined as the expected number of cases seroconverted, and the seroprevalence is obtained by dividing by the population size $N$. Formally, we project the seroprevalence for day $t \in T$ as

$$
\begin{aligned}
\pi^{(1)}(t) &= \frac{1}{N} \sum_i^{R(t+C)} \mathbb{E}[A_i(t) \mid \mathbf{y}^{Tan}] \\
&= \frac{1}{N} \sum_i^{R(t+C)} \hat{F}_U(t - (r_i - C)),
\end{aligned}
\tag{12}
$$

where $R(t + C)$ is the number of COVID-19 cases with symptom onset before day $t$. We

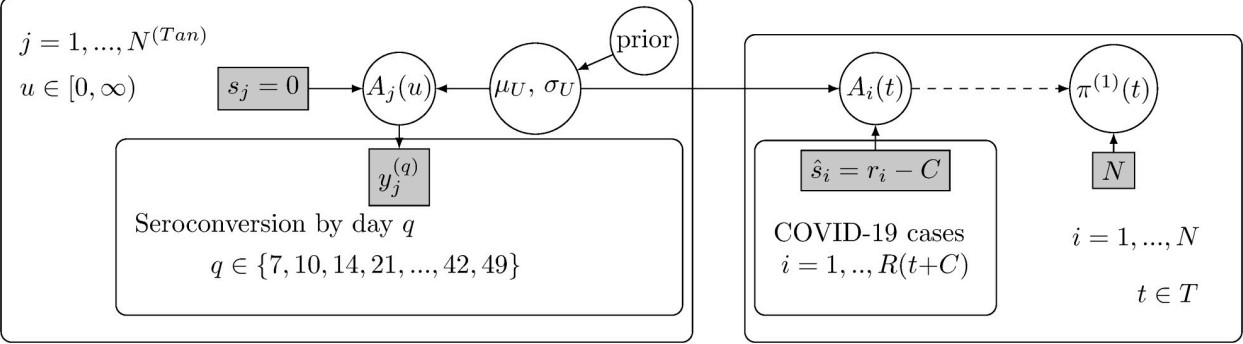

**Fig 6. The model for seroprevalence $\pi^{(1)}(t)$ (projection model).** Left plate: The duration from symptoms to seroconversion was modelled based on external data. Individuals $j$, $j = 1, \ldots, N^{(Tan)}$, experienced COVID-19 symptoms on day $s_j = 0$ and were tested for antibodies $q$ days later, where $q$ varied from 7 to 49 days. Individuals were tested on multiple days. Here, $A_j(u)$ denotes whether individual $j$ had seroconverted by day $u$, and $(y_j^{(q)})$ indicates the result of an antibody test taken on the $q$:th day. The duration from symptoms to seroconversion was modelled as a lognormal distribution with parameters $(\mu_U, \sigma_U)$. Right plate: The posterior distribution of $(\mu_U, \sigma_U)$ is utilised to project the seroconversion status $A_i(t)$ for each individual $i = 1, \ldots, R(t + C)$ with COVID-19 symptom onset before day $t \in T$ during the study period. The symptoms are assumed to have occurred on day $\hat{s}_i = r_i - C$, where $C$ is the lag from symptom onset to the COVID-19 diagnosis day $r_i$. The individual projections are used to derive the population level projection for the seroprevalence on day $t$, $\pi^{(1)}(t)$.

call $\pi^{(1)}(t)$ the projected seroprevalence. The Projection model is described graphically in Fig 6.

## Estimation of seroprevalence and underreporting

In the Estimation model, the posterior distribution for the parameter vector $g(\pi^{(0)})$ was obtained by sampling from $p(g(\pi^{(0)}) \mid \mathbf{y}^{(z)}; \Phi, \delta^{(z)})$, see Eq 6. Each sample was then transformed with $g^{-1}(.)$ to obtain samples from the posterior distribution of each weekly seroprevalence $\pi_w^{(0)}$. This provided samples for each day $t \in w$ of the week, resulting in samples from the posterior distribution of each daily seroprevalence $\pi^{(0)}(t)$, $t \in T$.

In the Projection model, the posterior distribution for $\theta$ was obtained by sampling from $p(\theta \mid \mathbf{y}^{Tan})$, see Eq 10. For each posterior sample and for each day $t \in T$ during the study period, seroprevalence was projected as described in Eq 12, resulting in samples from the posterior predictive distribution of each daily seroprevalence $\pi^{(1)}(t)$, $t \in T$.

Identical number of samples ($S = 40000$) were drawn from the posterior distributions of $\pi^{(0)}(t)$ and $\pi^{(1)}(t)$. For each sample from $\pi^{(0)}(t)$ and $\pi^{(1)}(t)$, a sample from $\Delta(t)$ was obtained by division, repeating over each day $t \in T$ during the study period.

We utilised the No-U-Turn Sampler algorithm for sampling, which is an efficient Markov Chain Monte Carlo algorithm [12]. We used STAN and the R package Rstan to carry out the sampling and monitored convergence via the Rhat statistic [13–15]. The STAN model code and an R code example are available on github.com/TuomoNieminen/covid19underreporting.

The choices for hyperparameters and other needed quantities to carry out the estimation are shown in S1 Table. See section Sensitivity analysis for sensitivity analysis regarding the hyperparameter choices.

## Ethics

The study protocol was approved by the ethical committee of the Hospital District of Helsinki and Uusimaa (HUS/1137/2020). Written informed consent was obtained from all participants.

## Results

### SARS-CoV-2 seroprevalence and the cumulative incidence of COVID-19

Table 2 shows the weekly numbers of blood samples and antibody test results in the serosurveys during the first epidemic wave. Out of 1465 samples taken between 9th April 2020 and 3rd July 2020, a total 35 (2.39%) were screening test positive and a total 7 (0.48%) were confirmation test positive. Five of the confirmed positive samples were taken before 4th May 2020, when the weekly numbers of samples were high, and they correspond to weekly sample seroprevalences 0.29%, 0.43% and 1.18%. After 4th May 2020, the weekly number of available samples decreased significantly and only two confirmed positive samples were observed.

Table 2 also shows the cumulative incidence of COVID-19 cases in the study population and in all HUS area residents. Three weeks prior to the first confirmed positive blood sample, the cumulative incidence of COVID-19 in the study population was 0.07% (736 cases, population 1.0 million), and in three weeks it increased to 0.13% (1330 cases). In all HUS area residents the cumulative incidence of COVID-19 was 0.31% by the end of the first epidemic wave (5348 cases, population 1.7 million).

**Table 2. COVID-19 cases and serology survey results in the Helsinki-Uusimaa region during spring 2020.**

| Period | COVID-19 cases[a] (cumulative) | | | Serological surveys[b] (weekly) | |
|---|---|---|---|---|---|
| | HUS (%) | Study (%) | Samples | Screening pos. (%) | Confirmation pos. (%) |
| 10.02.2020—16.02.2020 | 0–10 | 0–10 | - | - | - |
| 17.02.2020—23.02.2020 | 0–10 | 0–10 | - | - | - |
| 24.02.2020—01.03.2020 | 0–10 | 0–10 | - | - | - |
| 02.03.2020—08.03.2020 | 24 (0) | 20 (0) | - | - | - |
| 09.03.2020—15.03.2020 | 220 (0.01) | 190 (0.02) | - | - | - |
| 16.03.2020—22.03.2020 | 611 (0.04) | 505 (0.05) | - | - | - |
| 23.03.2020—29.03.2020 | 965 (0.06) | 737 (0.07) | - | - | - |
| 30.03.2020—05.04.2020 | 1578 (0.09) | 1030 (0.1) | - | - | - |
| 06.04.2020—12.04.2020 | 2212 (0.13) | 1332 (0.13) | 23 | 1 (4.35) | 0 (0) |
| 13.04.2020—19.04.2020 | 2825 (0.16) | 1621 (0.16) | 339 | 8 (2.36) | 1 (0.29) |
| 20.04.2020—26.04.2020 | 3436 (0.2) | 1895 (0.19) | 465 | 13 (2.8) | 2 (0.43) |
| 27.04.2020—03.05.2020 | 3965 (0.23) | 2138 (0.21) | 170 | 4 (2.35) | 2 (1.18) |
| 04.05.2020—10.05.2020 | 4466 (0.26) | 2415 (0.24) | 139 | 2 (1.44) | 0 (0) |
| 11.05.2020—17.05.2020 | 4804 (0.28) | 2636 (0.26) | 88 | 2 (2.27) | 1 (1.14) |
| 18.05.2020—24.05.2020 | 4987 (0.29) | 2747 (0.28) | 47 | 0 (0) | 0 (0) |
| 25.05.2020—31.05.2020 | 5118 (0.3) | 2825 (0.28) | 48 | 0 (0) | 0 (0) |
| 01.06.2020—07.06.2020 | 5200 (0.3) | 2863 (0.29) | 48 | 2 (4.17) | 1 (2.08) |
| 08.06.2020—14.06.2020 | 5240 (0.31) | 2885 (0.29) | 44 | 1 (2.27) | 0 (0) |
| 15.06.2020—21.06.2020 | 5279 (0.31) | 2899 (0.29) | 23 | 0 (0) | 0 (0) |
| 22.06.2020—28.06.2020 | 5315 (0.31) | 2915 (0.29) | 9 | 0 (0) | 0 (0) |
| 29.06.2020—04.07.2020 | 5347 (0.31) | 2932 (0.29) | 22 | 2 (9.09) | 0 (0) |
| All weeks | 5347 (0.31) | 2932 (0.29) | 1465 | 35 (2.39) | 7 (0.48) |

The column COVID-19 cases (cumulative) shows the cumulative number and cumulative incidence of COVID-19 cases by the end of each week (Period). The column Serological surveys (weekly) shows weekly results from the serological surveys for the target population of the current study.

[a] HUS: COVID-19 cases in the Helsinki-Uusimaa region of Finland; Study: COVID-19 cases in the target population of the current study. Populations 1.72M and 1.00M, respectively.

[b] Samples gives the weekly number of blood samples. Screening pos. (%) gives the weekly number and proportion of samples where SARS-CoV-2 IgG antibodies were detected with the screening test. Confirmation pos. (%) gives the weekly number and proportion of positive samples confirmed via a microneutralisation test.

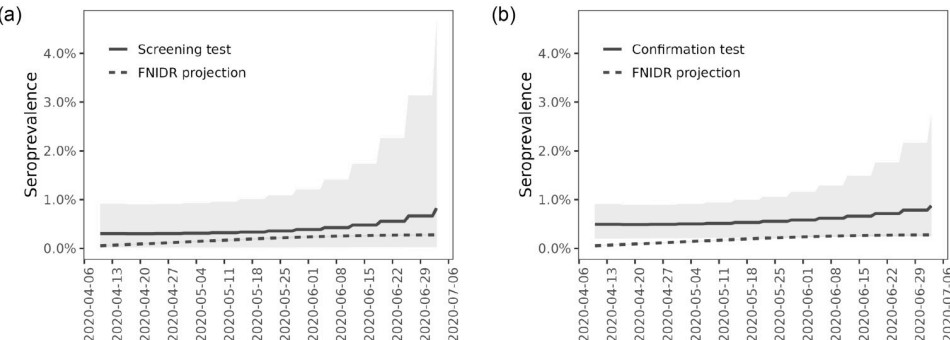

**Fig 7. Seroprevalence in the Helsinki-Uusimaa region during the first wave of the COVID-19 epidemic.** Both images show the posterior means and 95% credible intervals of seroprevalence $\pi^{(0)}(t)$ and projected seroprevalence $\pi^{(1)}(t)$, using the serology survey observations and COVID-19 cases (FNIDR projection), respectively. Solid lines are posterior means and the shaded areas correspond to 95% credible intervals derived from the pointwise 2.5% and 97.5% quantiles. The dashed lines show the projected seroprevalence and the shaded areas correspond to 95% credible intervals, however, the intervals for projected seroprevalence are very narrow and not visible. The image on the left shows results for the screening test and the image on the right shows results for the confirmation test. (a) Posterior means and 95% credible intervals of seropreva-lence $\pi^{(0)}(t)$ (Screening test) and projected seroprevalence $\pi^{(1)}(t)$. (b) Posterior means and 95% credible intervals of seroprevalence $\pi^{(0)}(t)$ (Confirmation test) and projected seroprevalence $\pi^{(1)}(t)$.

Fig 7 shows the estimates and projections of the seroprevalence, obtained under the Estimation model and Projection model. Results are shown for both the screening and confirmation tests. Based on the confirmation test, the posterior mean of the seroprevalence remains around 0.5% until the end of the study period where it slightly increases. The increase at the end is affected by the prior trend, combined with a low number of available blood samples. Based on the screening test, the seroprevalence behaves similarly but the posterior mean is lower and the posterior variance is greater. In both cases, the posterior mean of the projected seroprevalence (based on the COVID-19 cases) remains lower than the posterior mean of the estimated seroprevalence. The discrepancy to the estimated seroprevalence is greater during the beginning of the study period compared to the rest of the study period.

Table 3 shows the estimates and projections of the seroprevalence for selected dates during the study period. Based on the confirmation test, the estimated seroprevalence in the HUS area was 0.49 (95% CrI: 0.20–0.91) on 9th April 2020 and 0.58 (95% CrI: 0.23–1.16) on 28th May 2020. The corresponding seroprevalence projections based on COVID-19 cases are 0.06 (95% CrI:0.05–0.06) and 0.23 (95% CrI:0.21–0.24), respectively. Fig 8 shows the posterior distributions of the seroprevalence obtained under the Estimation model on 28th May 2020. Based on the confirmation test, the interquartile range (IQR) for the seroprevalence was 0.4%–0.67%. The seroprevalence based on the screening test has more uncertainty and the posterior median is lower.

## Underreporting

Fig 9 shows the posterior mean and quantiles of the underreporting ratio $\Delta(t)$ (see Eq 1), based on either the confirmation or the screening tests. For both tests, the posterior mean of $\Delta(t)$ first decreases, indicating higher underreporting during the beginning of the epidemic, then settles at around 2–3, and finally increased slightly toward the end of the first wave.

Table 3 shows the posterior mean and credible interval of $\Delta(t)$ for selected dates during the study period. Based on the confirmation test, there had been 8.9 (95% CrI: 3.6–16.5) infections for every COVID-19 case up until 9th April 2020. The underreporting then decreased, and up

**Table 3. Estimated and projected seroprevalences and the underreporting ratios during the study period.**

| date | COVID-19 cases $\pi^{(1)}(t)$ | Confirmation test $\pi^{(0)}(t)$ | $\Delta(t)$ | Screening test $\pi^{(0)}(t)$ | $\Delta(t)$ |
|---|---|---|---|---|---|
| 09.04.2020 | 0.055 (0.050–0.061) | 0.49 (0.20–0.91) | 8.92 (3.64–16.53) | 0.30 (0.022–0.91) | 5.49 (0.40–16.53) |
| 16.04.2020 | 0.080 (0.072–0.087) | 0.49 (0.20–0.89) | 6.14 (2.54–11.26) | 0.30 (0.022–0.90) | 3.79 (0.28–11.34) |
| 23.04.2020 | 0.106 (0.096–0.114) | 0.49 (0.21–0.89) | 4.68 (1.95– 8.53) | 0.30 (0.023–0.91) | 2.89 (0.21– 8.58) |
| 30.04.2020 | 0.132 (0.121–0.141) | 0.50 (0.21–0.91) | 3.81 (1.59– 6.95) | 0.31 (0.023–0.92) | 2.37 (0.17– 7.02) |
| 07.05.2020 | 0.158 (0.145–0.168) | 0.51 (0.22–0.94) | 3.27 (1.37– 6.00) | 0.32 (0.024–0.96) | 2.05 (0.15– 6.12) |
| 14.05.2020 | 0.184 (0.170–0.194) | 0.53 (0.22–0.99) | 2.90 (1.20– 5.43) | 0.34 (0.024–1.01) | 1.84 (0.13– 5.52) |
| 21.05.2020 | 0.209 (0.194–0.220) | 0.56 (0.23–1.06) | 2.67 (1.08– 5.11) | 0.36 (0.025–1.09) | 1.72 (0.12– 5.24) |
| 28.05.2020 | 0.230 (0.214–0.241) | 0.58 (0.23–1.16) | 2.54 (1.01– 5.04) | 0.39 (0.027–1.21) | 1.69 (0.12– 5.28) |
| 04.06.2020 | 0.246 (0.231–0.257) | 0.62 (0.24–1.29) | 2.51 (0.96– 5.27) | 0.43 (0.028–1.41) | 1.73 (0.11– 5.74) |
| 11.06.2020 | 0.259 (0.244–0.268) | 0.66 (0.24–1.49) | 2.56 (0.93– 5.75) | 0.48 (0.029–1.73) | 1.86 (0.11– 6.69) |
| 18.06.2020 | 0.268 (0.254–0.276) | 0.71 (0.25–1.76) | 2.67 (0.92– 6.57) | 0.56 (0.031–2.26) | 2.08 (0.11– 8.50) |
| 25.06.2020 | 0.274 (0.262–0.281) | 0.78 (0.25–2.16) | 2.86 (0.91– 7.91) | 0.67 (0.032–3.14) | 2.43 (0.12–11.53) |
| 02.07.2020 | 0.279 (0.268–0.285) | 0.88 (0.25–2.73) | 3.14 (0.91– 9.80) | 0.83 (0.033–4.66) | 2.97 (0.12–16.71) |

The column COVID-19 cases shows the projected seroprevalence $\pi^{(1)}(t)$, and the columns Confirmation test and Screening test show estimated seroprevalence $\pi^{(0)}(t)$, and the underreporting ratio ($\Delta(t)$), see Eq 1) of SARS-CoV-2 infections during the study period. The seroprevalences are shown in percentage scale. Displayed are the posterior means along with 95% credible intervals, derived from the 2.5% and 97.5% quantiles of the distributions.

until 28th May 2020 our estimate is that there had been 2.5 (95% CrI: 1.0–5.0) SARS-CoV-2 infections for every COVID-19 case. The estimate of the underreporting ratio then remained at the same level until the end of the first wave.

Fig 10 shows the posterior distribution for the underreporting ratio by 28th May 2020, based on either the screening or confirmation tests. Based on the confirmation test, the IQR for underreporting was 1.8—3.0. The estimate derived from the screening test data has more uncertainty and shows lower underreporting.

## Time from COVID-19 symptom onset to seroconversion

S4 Fig describes the posterior distributions of $\mu_U$ and $\sigma_U$, the parameters of the lognormal distribution of the time from COVID-19 symptom onset to seroconversion. The posterior

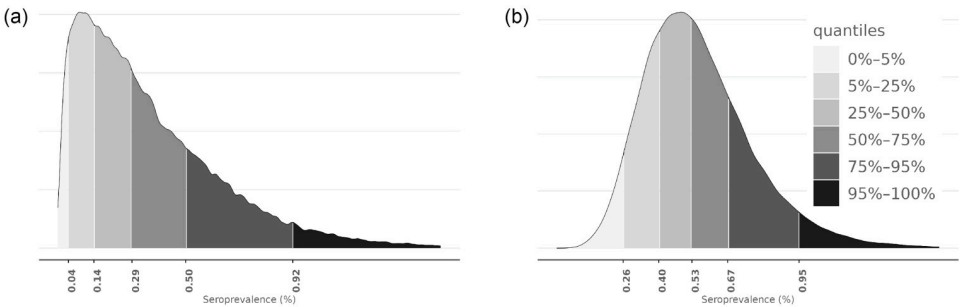

**Fig 8. Posterior distributions of seroprevalence $\pi^{(0)}(t)$.** In the images, $t$ corresponds to 8th May 2020, learned from the screening (left image) and confirmation (right image) test data. The seroprevalence is shown in percentage scale. (a) Posterior density of $\pi^{(0)}(t)$, where $t$ corresponds to 8th May 2020 (Screening test). (b) Posterior density of $\pi^{(0)}(t)$, where $t$ corresponds to 8th May 2020 (Confirmation test).

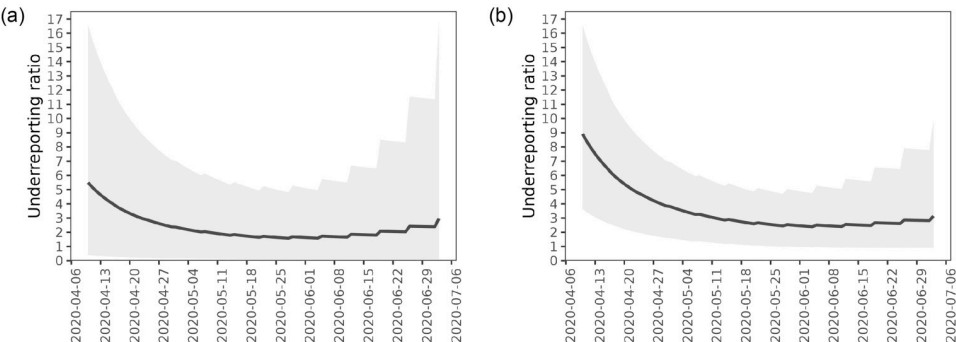

**Fig 9. Extent of underreporting in the Helsinki-Uusimaa region during the first wave of the COVID-19 epidemic.** Both figures show estimates for the underreporting ratio $\Delta(t)$. Solid lines are posterior means and the shaded areas correspond to 95% credible intervals derived from the pointwise 2.5% and 97.5% quantiles. The image on the left shows results for the screening test and the image on the right shows results for the confirmation test. (a) Posterior means and 95% credible intervals of $\Delta(t)$ (Screening test). (b) Posterior means and 95% credible intervals of $\Delta(t)$ (Confirmation test).

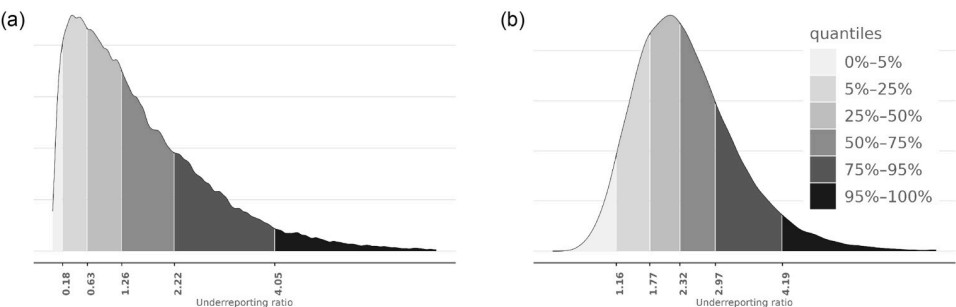

**Fig 10. Posterior distributions of underreporting ratio $\Delta(t)$.** In the images, $t$ corresponds to 8th May 2020, learned from the screening test (left image) and confirmation test (right image) data. (a) Posterior density of $\Delta(t)$, where $t$ corresponds to 8th May 2020 (Screening test). (b) Posterior density $\Delta(t)$, where $t$ corresponds to 8th May 2020 (Confirmation test).

medians of $\mu_U$ and $\sigma_U$ are 2.87 and 0.72, respectively. The figure also shows the posterior predictive distribution for the time from symptom onset to seroconversion. The predicted median delay from symptom onset to seroconversion is close to 18 days and the 75% quantile is over 29 days. By day 60 since symptom onset, the probability of seroconversion is over 95%.

## Sensitivity analysis

S5 Fig shows the prior and posterior distributions of $\sigma$, the strength of dependency in the Estimation model, learned from the screening and confirmation test data. In both cases, the posterior distribution is similar to the (informative) prior distribution, indicating that the data do not contain much information about $\sigma$ and the analysis may be sensitive to the prior distribution of $\sigma$.

S2 Table shows estimates of the underreporting ratio $\Delta(t)$, based on data from the confirmation test, using different values for hyperparameters $\mu_1$, $\sigma_1$ and $\beta$. Smaller values of $\beta$ correspond to a higher prior expectation and higher prior variance for $\sigma$ and in turn higher posterior variance for $\Delta(t)$. Smaller values of $\beta$ also correspond to slightly higher posterior

means for $\Delta(t)$, but only at the start and end of the study. For example, comparing choices $\beta = 2$ to $\beta = 40$ when $\alpha = 2$, $\text{logit}(\mu_1) = 0.05$ and $\sigma_1 = 2$, the 95% credible intervals for $\Delta(t)$ on 28th May 2020 are $0.4 - 7.3$ and $1.0 - 5.0$, respectively, however, the posterior means are almost identical (2.54 and 2.56). With a choice of $\beta = 120$, the underreporting ratio estimates are similar on 28th May 2020 and at the end of the study, while with smaller values of $\beta$ there is more uncertainty in the estimates at the end of the study.

Choice of a larger $\text{logit}(\mu_1)$ corresponds to a higher posterior mean for $\Delta(t)$, but only marginally. For example, comparing the choice $\text{logit}(\mu_1) = 0.005$ to $\text{logit}(\mu_1) = 0.05$ when $\sigma_1 = 2$, $\beta = 40$, the posterior means for $\Delta(t)$ on 28th May 2020 are 2.4 and 2.6, respectively. A choice of smaller $\sigma_1$ reduces the posterior variance of $\Delta(t)$ and elevates the effect of the chosen $\mu_1$, but the effects are small.

In all cases, the effects of the hyperparameter choices are magnified towards the end of the study period, when the number of available samples from the serosurveys is low.

## Discussion

We estimated that with 95% probability there were 1 to 5 SARS-CoV-2 infections for every COVID-19 case during the first epidemic wave in Finland. A 50% probability interval for the underreporting was 1.8–3.0. The underreporting was highest before April 2020 when there were 4 to 17 infections for every COVID-19 case (95% probability). It is likely that the seroprevalence in the Helsinki-Uusimaa region was over 0.5% already by the end of May 2020 (95% CrI: 0.2–1.2), while the cumulative incidence of COVID-19 cases in the region was 0.3% by the end of June. Based on the estimate of underreporting and the cumulative incidence of COVID-19 cases, we estimate that between 0.5%–1% (50% probability) and no more than 1.5% (95% probability) of the population in the Helsinki-Uusimaa region were infected with SARS-CoV-2 by the end of June 2020.

There is great uncertainty about the estimated seroprevalence and the corresponding estimate of underreporting at the end of the study period, due to the small number of samples available in the serosurveys. The estimates are therefore sensitive to the model specification (i.e. hyperparameters). Accordingly, we consider the most robust estimate of underreporting during the first wave pertaining to the end of May 2020. We do not expect that the magnitude of underreporting changed significantly during the rest of the first wave, as there were no changes in virus testing policy or capacity. While our analysis included prior information related to the dependency between seroprevalences on consecutive weeks, our sensitivity analysis indicated that a prior choice of stronger dependency could result in more robust estimates of underreporting towards the end of the first wave.

Our analysis leaves a small but reasonable probability that by the end of the first wave there was no underreporting at all. Our estimation approach allowed values of the underreporting ratio below one, which would correspond to there being more COVID-19 cases than SARS-CoV-2 infections. This could occur in theory, in case the diagnosis procedure for COVID-19 (i.e. PCR test) was unspecific and the virus testing was widespread. Nevertheless, we believe this to be unrealistic in our study and simply interpret values below one to represent absence of underreporting. It seems, however, also unrealistic that no underreporting occurred, as in the general population the virus testing was targeted to symptomatic individuals only. Findings from a population-based screening in Iceland during March 2020 show that 43% of individuals who tested positive for SARS-CoV-2 were asymptomatic and findings from Spain indicate that one third of infections were asymptomatic in April-May 2020 [3, 16]. A systematic review and meta-analysis of 95 published studies estimates that globally 41% (34%—48%) of confirmed COVID-19 cases were asymptomatic during the pre COVID-19-vaccine

era [17]. Our analysis also leaves a small but reasonable probability that only 20% or less of SARS-CoV-2 infections were detected during the first epidemic wave. We believe that this may still be plausible, as other countries show even higher underreporting [3, 5].

A key assumption in our analysis was that the serosurvey participants represented the population of interest. The participation rate was 50%–64% and there were several factors which may have caused selection bias, as survey participation may correlate with the likelihood of SARS-CoV-2 infection. First, during the first two weeks, the surveys targeted only few of the largest municipalities. These had the highest numbers of COVID-19 cases, which may lead to overestimating the seroprevalence and thus the underreporting. However, an analysis using data only from the largest municipality (Helsinki) showed similar estimates of underreporting. Secondly, the participation rate in younger age groups (18–29) was lower than in other age groups. Age is likely associated with the incidence of SARS-CoV-2 infection due to differences in social behaviour. In April 2020, Finns aged 18–29 had a similar frequency of daily social contacts than those aged 30–59, but a higher frequency of contacts than those aged 60–69 [18]. The underrepresentation of young adults in our study can lead to underestimation of the seroprevalence and of underreporting. Third, in several population health examination surveys, participation rates have been found lower among individuals with lower education [19]. Those individuals often work in professions where working remotely and social distancing may be more difficult to arrange, and thus they may be more exposed to infection. If those previous findings hold in this survey, this can lead to underestimation of seroprevalence and thus underreporting. Fourth, historically, the participation rate in Finnish health examination surveys has been lower in language groups other than Finnish and Swedish [20]. The incidence of COVID-19 during the first epidemic wave was several times higher in language groups other than Finnish, Swedish, English or Russian (S6 Fig). However, as the target population of our study includes only those four language groups, we do not believe that the possible underrepresentation of language groups other than Finnish and Swedish is likely to bias our results. Finally, our preliminary analyses from the serosurveys beyond the first wave indicate that subjects with a past confirmed SARS-CoV-2 infection tend to have a lower participation rate. It is possible that those with a confirmed infection were less willing to participate. This can lead to underestimation of the seroprevalence and of underreporting.

Our study was limited to those 18–69 years old. For ethical reasons, the elderly most vulnerable to severe COVID-19 were not invited to participate during the beginning of the epidemic as participation required a medical site visit and therefore could increase the risk of infection with SARS-CoV-2. Children were not invited due to difficulties in obtaining informed consent from minors. The median age of COVID-19 cases in the HUS area showed a decreasing trend during spring 2020, most likely due to the increase in testing capacity, allowing detection of milder disease cases (S7 Fig). It is therefore likely that the underreporting was both higher and decreased more in the younger age groups during the first epidemic wave. Other serological studies have used regression analysis or post stratification to account for differences in the age and sex distributions between the survey participants and the underlying population [3, 4, 21]. These methods could help reduce bias and allow for the estimation of age-dependent underreporting. We decided not to use such analytical methods due to the very small number of confirmed positive samples.

Another key assumption in our analysis was that the time-dependent probability of seroconversion after COVID-19 symptoms, as estimated from the external data set from Tan et al., is similar to how the antibody detection in the serological surveys would perform. Otherwise, the underreporting ratio, i.e. the ratio of the estimated seroprevalence (based on serosurveys) and the projected seroprevalence (based on COVID-19 cases) may not

accurately describe underreporting. The patients in Tan et al. were all hospitalised and several of them were classified with severe pneumonia. By contrast, the majority of the FNIDR COVID-19 cases during the first epidemic wave did not require hospital care. Severe cases may have higher antibody responses, and this may cause us to overestimate the projected seroprevalence and hence underestimate the underreporting [22]. Additionally, the SARS-COV-2 antibody detection method utilised in Tan et al. differed from the methods utilised in the serosurveys. The serosurvey antibody detection was calibrated to be 100% sensitive by day 30 since infection. By contrast, in Tan et al., only 74% of the patients had seroconverted by day 28 since symptom onset, and accordingly, our seroprevalence projection yielded approximately 75% probability of seroconversion by day 30 since symptom onset. This discrepancy indicates that we may overestimate the time lag to developing detectable antibodies after COVID-19 symptoms. This in turn indicates that we may overestimate the underreporting during the beginning of the epidemic, at worst by a factor of around 0.75. Therefore, instead of 4–17 there were perhaps 3–13 infections for every COVID-19 case before April.

When projecting the seroprevalence, we assumed that the probability of seropositivity following COVID-19 symptoms is strictly increasing over time. In reality, the antibody levels eventually wane and after 8 months since SARS-CoV-2 infection, the N-IgG antibodies are detectable in only 66% of individuals [22]. Our analysis covers a period of four months, and there were not many infections in Finland before March 2020, so at worst we measured antibodies from serosurvey participants who were infected four months ago. The detectability of antibodies would then be at least 66% and possibly over 80%, assuming a linear decrease from the 100% detectability at one month. By contrast, our seroprevalence projection gives an almost 100% probability of seropositivity at four months since COVID-19 symptom onset. This worst-case discrepancy would correspond to overestimating the underreporting by 20% at the end of the study period. To analyse data beyond the first epidemic wave, the seroprevalence projection should be modified to allow for a decrease in the probability of seropositivity after appropriate time.

We included an analysis based on the screening test to demonstrate how our method can be used with tests which are not fully specific. The estimates of seroprevalence based on the screening test were lower than those based on the confirmation test, when adjusting for the expected false positive rate of the screening test. This implies that either the specificity of the screening test was higher than expected, or alternatively, the confirmation test was not fully specific. The confirmation test utilises a microneutralisation test (MNT) as the second test to confirm the presence of SARS-CoV-2 antibodies. Based on an analysis of a large number of pre-pandemic sera from different age cohorts, the MNT can be considered to be fully specific [10]. It is therefore extremely unlikely that any of our 7 confirmed positives samples was a false positive; more likely the true specificity of the screening test was higher than we assumed. In our analysis, we assumed that the specificity of the screening test was a known constant, based on a 81/83 true negative finding. In reality, however, there is uncertainty in the exact specificity, and the results derived from the screening test therefore have more uncertainty than our analysis implies. For analysing data from a test with unknown specificity, we agree with treating the specificity as an unknown parameter, as recommended by Gelman and Carpenter, and as implemented by e.g. Stringhini et al. [4, 23].

We used a constant value 3.5 days as the delay from symptom onset to COVID-19 diagnosis, which was based on expert evaluation and information available during early 2020. In reality, the exact delay is unknown and subject to variation. It is likely that 3.5 days is a reasonable estimate for the average delay in the Helsinki-Uusimaa region during spring 2020, as according to internal infection tracking data at our institute, in the capital city (Helsinki) the delay

was close to 6 days in March 2020, close to 4 days in April and close to 3 days in May. Small variations in this delay do not affect our analysis, as small changes in the COVID-19 symptom onset day would not significantly alter the seroprevalence projection. Our results are therefore not sensitive to small changes in the choice of delay.

Our results imply that the spread of SARS-CoV-2 infection was very limited in Finland during spring 2020 compared to other European countries, as seroprevalence was still likely under 1% in the densely populated Helsinki-Uusimaa region by the beginning of June. For example in Spain seroprevalence was likely over 10% around Madrid by May 11 [3], and in Geneva, Switzerland, it was 10.8% (8.2–13.9) by May 9th. Finland had the advantage of being slightly isolated from main land Europe and therefore the epidemic started a few weeks later, giving more time to implement social distancing. The general public's compliance with epidemic recommendations was likely very high. There was a large reduction in the daily numbers of social contacts in the early part of the 2020 COVID-19 epidemic in Finland, which was likely a major contributor to the steady decline of the epidemic in the country [18].

In summary, we presented a Bayesian approach to estimate the time-dependent underreporting of SARS-CoV-2 infections during the COVID-19 epidemic. We implemented the proposed approach to data from adults living in the Helsinki-Uusimaa region of Finland during the first epidemic wave in 2020. The analysis we here describe can also be applied in real time, and our method informed about the spread, detection, and severity of SARS-CoV-2 infection in Finland during 2020. Our results indicate that most SARS-CoV-2 infections were not detected and the underreporting was most severe during the beginning of the epidemic. However, as the cumulative incidence of COVID-19 was very low, it is likely that less than 1.5% of the population in the Helsinki-Uusimaa region had been infected with SARS-CoV-2 by the beginning of July 2020. Assuming that the underreporting was similar in other parts of the country and in children and the elderly, the first wave of the COVID-19 epidemic left a vast majority of the Finnish population unaffected, with almost the entire population still unexposed and susceptible to SARS-CoV-2.

## Supporting information

**S1 Table. Parameters of the prior distribution in the estimation model, and the specificities of the screening and confirmation tests.**
(PDF)

**S2 Table. Influence of choices of hyperparameters on the estimation of underreporting ratio $\Delta(t)$.** Shown are posterior means and 95% credible intervals for $\Delta(t)$, based on the confirmation test data, for 9th April 2020 ($t = 0$), 28th May 2020 ($t = 49$) and 2nd July 2020 ($t = 84$), using different values for the parameters $\mu_1$, $\sigma_1$, and $\beta$. The value used for the parameter $\alpha$ was 2.
(PDF)

**S1 Fig. Age distributions of study sub-populations.** Age distributions of: population in the Helsinki-Uusimaa region at the end of 2021 (HUS); COVID-19 cases for the HUS population during the first wave of the COVID-19 epidemic in 2020 (FNIDR); the study population, i.e. the target population of the current study (HUS (incl.)); COVID-19 cases from the study population during the first wave of the COVID-19 epidemic in 2020 (FNIDR (incl.)); serological survey participants from the study population during the first wave (Serosurveys).
(TIF)

**S2 Fig. The serological survey antibody tests and their performances on the calibration data.** The screening test is the result of the IgG antibody test, which may give false positive

results. The confirmation test is a combination of the IgG and microneutralization tests (MNT), where the IgG positive samples are tested again with the MNT. After optimizing performance on the calibration data, which includes samples from PCR positive and negative individuals, the sensitivity and specificity of the screening test are 33/33 (100%) and 81/83 (97.59%), respectively, while the sensitivity and specificity of the confirmation test are 33/33 (100%) and 83/83 (100%), respectively.
(TIF)

**S3 Fig. Estimation model seroprevalence prior distribution.** Prior mean, and 2.5% and 97.5% quantiles for each weekly seroprevalence $\pi_w^{(0)}$ in the Estimation model. The estimates were computed based on 40000 samples generated from the prior distribution of $\pi$.
(TIF)

**S4 Fig. Time from COVID-19 symptom onset to seroconversion.** The three images show, starting from the the left: the posterior distribution for $\mu_U$, the posterior distribution for $\sigma_U$, and the posterior predictive distribution for $U$, the time from COVID-19 symptom onset to seroconversion. The distribution for $U$ was obtained by sampling from the lognormal distribution, using samples from the joint posterior distribution for $(\mu_U, \sigma_U)$.
(TIF)

**S5 Fig. Prior and posterior distributions for the parameter $\sigma$.** Image on the left shows the prior distribution, the middle image shows the posterior distribution based on confirmation test data, and the image on the right shows the posterior distribution based on the screening test data.
(TIF)

**S6 Fig. Incidence of COVID-19 cases in the Helsinki-Uusimaa region by age group and language during the first wave of the epidemic in 2020.** The language groups are Finnish (fi), Swedish (sv), English (en), Russian (ru) and other.
(TIF)

**S7 Fig. Age distribution of COVID-19 cases in the Helsinki-Uusimaa region during the first wave of the COVID-19 epidemic in 2020.**
(TIF)

## Acknowledgments

We thankfully acknowledge the fluent collaboration with the Digital and Population Data Services Agency DVV for access to the Finnish Population Information System (PIS) and especially for HUS Diagnostic Center HUSLAB for study sample logistics. We thank all the study participants. We thank Juha Oksanen, Esa Ruokokoski, Elina Isosaari, Niina Ikonen, Dennis Ahlfors, Timo Koskenniemi, Nina Ekström, Pamela Österlund, Anu Haveri and Camilla Virta for their contributions related to data management and analyses.

## Author Contributions

**Conceptualization:** Kari Auranen, Sangita Kulathinal, Merit Melin, Arto A. Palmu, Jukka Jokinen.

**Data curation:** Tuomo A. Nieminen, Merit Melin, Arto A. Palmu.

**Formal analysis:** Tuomo A. Nieminen.

**Funding acquisition:** Merit Melin, Arto A. Palmu, Jukka Jokinen.

**Investigation:** Merit Melin, Arto A. Palmu.

**Methodology:** Tuomo A. Nieminen, Kari Auranen, Sangita Kulathinal, Tommi Härkänen, Jukka Jokinen.

**Project administration:** Merit Melin, Arto A. Palmu, Jukka Jokinen.

**Resources:** Merit Melin, Arto A. Palmu, Jukka Jokinen.

**Software:** Tuomo A. Nieminen.

**Supervision:** Kari Auranen, Sangita Kulathinal, Merit Melin, Arto A. Palmu, Jukka Jokinen.

**Visualization:** Tuomo A. Nieminen.

**Writing – original draft:** Tuomo A. Nieminen, Kari Auranen, Sangita Kulathinal.

**Writing – review & editing:** Tuomo A. Nieminen, Kari Auranen, Sangita Kulathinal, Tommi Härkänen, Merit Melin, Arto A. Palmu, Jukka Jokinen.

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
