## [Decision Letter · Decision Letter 0]

21 Mar 2023

PONE-D-22-33699Underreporting of SARS-CoV-2 infections during the first wave of the 2020 COVID-19 epidemic in Finland - Bayesian inference based on a series of serological surveysPLOS ONE

Dear Dr. Nieminen,

Thank you for submitting your manuscript to PLOS ONE. After careful consideration, we feel that it has merit but does not fully meet PLOS ONE’s publication criteria as it currently stands. Therefore, we invite you to submit a revised version of the manuscript that addresses the points raised during the review process. In your revised manuscript, please pay particular attention to the issues raised by Reviewer 3 regarding the model being poorly estimated. Also note that although Reviewer 2 raises concerns about the novelty of the approach, PLOS One manuscripts are evaluated on the basis of methodological rigor and high ethical standards, regardless of perceived novelty.  

We look forward to receiving your revised manuscript.

Kind regards,

Timothy J Wade, Ph.D

Academic Editor

PLOS ONE

Journal Requirements:

2. We note that your ethics statement indicates "The study protocol was submitted for ethical review to the ethical review board of the Hospital District of Helsinki and Uusimaa. Written informed consent was obtained from all participants of the serological surveys." To ensure that your submission complies with our policy on human subject research (https://journals.plos.org/plosone/s/human-subjects-research) please clarify in the methods section of the manuscripts whether the ethical review board of the Hospital District of Helsinki and Uusimaa approved this study. If applicable please provide approval numbers.

“This study was funded by the Finnish Institute for Health and Welfare.”

“We report no conflict of interests related to the current work. Finnish Institute for Health and Welfare (THL) conducts Public-Private Partnership with vaccine manufacturers and has received research funding from Sanofi Inc., Pfizer Inc., and GlaxoSmithKline Biologicals SA for studies not related to COVID-19. Nieminen, Melin, Palmu and Jokinen have been investigators in these studies, but they have received no personal remuneration.”

7. Your ethics statement should only appear in the Methods section of your manuscript. If your ethics statement is written in any section besides the Methods, please move it to the Methods section and delete it from any other section. Please ensure that your ethics statement is included in your manuscript, as the ethics statement entered into the online submission form will not be published alongside your manuscript.

8. We note that Figures 1 and 2 in your submission contain map images which may be copyrighted. All PLOS content is published under the Creative Commons Attribution License (CC BY 4.0), which means that the manuscript, images, and Supporting Information files will be freely available online, and any third party is permitted to access, download, copy, distribute, and use these materials in any way, even commercially, with proper attribution. For these reasons, we cannot publish previously copyrighted maps or satellite images created using proprietary data, such as Google software (Google Maps, Street View, and Earth). For more information, see our copyright guidelines: http://journals.plos.org/plosone/s/licenses-and-copyright.

 a. You may seek permission from the original copyright holder of Figures 1 and 2 to publish the content specifically under the CC BY 4.0 license. 

Reviewers' comments:

Reviewer's Responses to Questions

**Comments to the Author**

1. Is the manuscript technically sound, and do the data support the conclusions?

Reviewer #1: Yes

Reviewer #2: No

Reviewer #3: Yes

Reviewer #4: Yes

2. Has the statistical analysis been performed appropriately and rigorously? 

Reviewer #1: Yes

Reviewer #2: Yes

Reviewer #3: Yes

Reviewer #4: Yes

3. Have the authors made all data underlying the findings in their manuscript fully available?

Reviewer #1: Yes

Reviewer #2: Yes

Reviewer #3: Yes

Reviewer #4: Yes

4. Is the manuscript presented in an intelligible fashion and written in standard English?

Reviewer #1: Yes

Reviewer #2: Yes

Reviewer #3: Yes

Reviewer #4: Yes

5. Review Comments to the Author

Reviewer #1: To assess under-reporting, the authors compared SARS-CoV-2 infections from serological surveys to reported COVID-19 illnesses during the first pandemic wave in Finland. Their analyses are rigorous, clearly described, and results are comparable to others (e.g., JAMA 2021; 326:1400-09, Lancet Reg Health Am 2023; 18:100403). As severity increases with co-morbidities, for which age is a proxy, and reporting increases with severity, I imagine that under-reporting of childhood infections was disproportionate. Could the authors stratify by age?

Reviewer #2: Review of “Underreporting of SARS-CoV-2 infections during the first wave of the 2020 COVID-19 epidemic in Finland – Bayesian inference based on a series of serological surveys”

by Tuomo A. Nieminen, Kari Auranen, Sangita Kulathinal, Tommi Härkänen, Merit Melin, Arto A. Palmu, Jukka Jokinen

Summary

In this manuscript, a Bayesian approach is used to estimate the underreporting of SARS-CoV-2 infections during the first wave of COVID-19 in Finland, that is, from March to June 2020. The analysis is based on a series of serological surveys.

It is estimated by the authors that there were 1 to 5 infections for every detected case during the first wave. Reporting is estimated to have been much poorer during the first months of this period (before April) with 4 to 17 infections for every detected case.

General comments

To estimate the underreporting of a disease is particularly important when the number of asymptomatic cases is high, as it is known to be the case for the COVID-19 disease. To do so, to use a series of serological surveys, when available as it is the case here, is a proper choice and to perform the analyses based on a Bayesian inference method appears also well designed for this purpose.

However, although the paper is well written and the analysis well driven, I found the results of rather limited reach. Maybe this methodology was not applied yet to COVID-19 in Finland, but it is rather common in itself (see e.g. [1-2]), and it has been applied to many other countries by other authors since the beginning of the pandemic (e.g. [3-5]). From our point of view, methodologically speaking, the approach is not sufficiently new to deserve a publication in PLoS.

From an epidemiological point of view, these results are nice and of some interest, but to make it really useful, one would expect to have these results put in perspective with other factors and/or with explanations on the behaviors specifically observed.

In particular, results appear rather different from what was observed in other countries in Europe but the present analysis does not help to understand these differences. For instance, the prevalence in Finland is significantly lower than most of the other European countries. Can the underestimation contribute to explain such a behavior? Or may it result from dynamical reasons (a model was recently obtained showing that, some epidemiological systems can have a very different time evolution in amplitude under strictly the same sanitary conditions [6]) or due to specific policies (as it has been the case in several Asian countries)?

Here, a retrospective analysis is performed by the authors based on a serological survey, but such a serological was not available at the very beginning of the epidemic. To cope with this difficulty, other authors have used more basic approaches based on case fatality ratio [7]. What would have been the effect of such a rough approach in comparison to the (more robust) Bayesian approach here used?

I think these types of questions will deepen the investigations and make the discussions and the work interesting to a wider audience.

Despite its technical interest and quality, at this stage, I don’t think the present work can help much to understand the behavior observed in Finland in comparison to other countries in Europe or in the world. For this reason, I cannot recommend it for publication in a PLoS journal.

References

[1] M. Dvorzak and H. Wagner, Sparse Bayesian modelling of underreported count data, Statistical Modelling, 2016, 16, 24-46.

[2] Turbé H, Bjelogrlic M, Robert A, Gaudet-Blavignac C, Goldman JP, Lovis C. Adaptive Time-Dependent Priors and Bayesian Inference to Evaluate SARS-CoV-2 Public Health Measures Validated on 31 Countries. Front Public Health, 2021, 8, 583401.

[3] Lope DJ, Demirhan H. 2022. Spatiotemporal Bayesian estimation of the number of under-reported COVID-19 cases in Victoria Australia. PeerJ, 10, e14184 http://doi.org/10.7717/peerj.14184

[4] Paixão B, Baroni L, Pedroso M, Salles R, Escobar L, de Sousa C, de Freitas Saldanha R, Soares J, Coutinho R, Porto F, Ogasawara E. Estimation of COVID-19 Under-Reporting in the Brazilian States Through SARI. New Gener Comput., 2021, 39(3-4), 623-645.

[5] Ricardo Cao & José E. Chacón (2022) Introduction to the special issue on Data Science for COVID-19, Journal of Nonparametric Statistics, 34(3), 555-569.

[6] Thenon N, Peyre M, Huc M, Touré A, Roger F, Mangiarotti S (2022) COVID-19 in Africa: Underreporting, demographic effect, chaotic dynamics, and mitigation strategy impact. PLoS Negl. Trop. Dis., 16(9), e0010735.

[7] Russell Timothy W , Hellewell Joel , Jarvis Christopher I , van Zandvoort Kevin , Abbott Sam , Ratnayake Ruwan , CMMID COVID-19 working group , Flasche Stefan, Eggo Rosalind M , Edmunds W John , Kucharski Adam J . Estimating the infection and case fatality ratio for coronavirus disease (COVID-19) using age-adjusted data from the outbreak on the Diamond Princess cruise ship, 2020. Euro Surveill. 2020, 25(12), 2000256. https://doi.org/10.2807/1560-7917.

Detailed comments

*p.2 “If the virus causes clinical disease, the rate of […]”: Maybe I would say instead “If the virus causes specific clinical disease, the rate of […]”?

*p.3 line 26 “the numbers of COVID-19 cases”: To avoid any misunderstanding I’d say “the numbers of COVID-19 new cases”

*p.4 line 1: Ref. [2] has been published now, since more than one year now… the authors did not update their bibliography before submitting the manuscript. See https://doi.org/10.1016/j.ijid.2020.12.038

*p.4 line 53-60: Indeed, it is important to have information from other countries for comparison.

*p.4 line 57-58 “[…] estimated that there were 11 SARS-Cov-2 infections for every COVID-19 case.”: Maybe I’d say “detected cases” instead of just “case”.

*p.4, line 73: Here also, the Ref. [6] was not updated. See 10.1172/JCI138759

*lines 72-85: it seems it is here that you explain what will be investigated in the present study. But it is not very clear, we have to deduce it by reading the two paragraphs. I think, you should state it more clearly, maybe with “in this paper” at line 72 “To better address the delays in antibody responses, in this paper, we utilise […] »

*line 81-83 “The novelty of our methodology is in accounting for the uncertainty in the time lag from disease symptoms to seroconversion when estimating the time-evolving underreporting of infections. Our analysis shows how the underreporting of SARS-CoV-2 infections evolved over time during the first epidemic wave in Finland.”: It appears to be the main contribution of this paper. I find it rather narrow for an international publication.

* lines 98-100 “These data consist of COVID-19 cases notified as either a positive SARS-CoV-2 finding from a microbiological laboratory or a clinical diagnosis by a medical doctor.”: If the two sources of information are separated, the analysis could be performed on the two datasets to investigate the robustness of the analysis.

Reviewer #3: General comments

This paper focusses on the underreporting of SARS-CoV-2 during the first wave of the 2020 epidemic in Finland. It focusses on Bayesian inference based on serological surveys. It uses different data sources to identify infection rates.

The paper is well written and clear mostly. The method is applied to data for age group 18 – 69 but doesn’t included older people. Older people ,possibly in care homes, could have a higher risk of spread or infection. There must be a greater explanation/justification for the exclusion.

Page 4 what is the extended capital region? Its not defined anywhere.

The focus is the estimation of the underreporting ratio which is a function of seroprevalence divided by case count. The seroprevalence is assumed to have a random walk prior distribution in the logit of the prevalence, and the count of tests is used to estimate the posterior distribution of the seroprevalence; whereas the case count is used in a binary model and the underlying

Table 3 displays the results of estimation for the seroprevalence from surveys and case counts.

There is an issue about this table however. A credible intervals is shown for the ratio (underreporting) but the two data streams are modelled separately. How can an interval be constructed for the ratio when these models are separately run using MCMC. The samples cant be shared.

An interpretation issue: according to Table 3 while the ratio stabilizes from June onwards (around 2.5) the credible interval crosses 1.0 and so the underreporting is poorly estimated. This is not mentioned but is a serious problem.

In general for a ratio to be estimated with a credible interval the ratio should have been computed within a joint model for seroprevalence and case counts.

Finally I note that various prior parameters are assumed for the distributions included in the models and these are given in Table S1 for the estimation model, and while Table S2 shows effects of varying some of these, it is noticeable that at later time the credible interval for the ratio crosses 1.0 for most entries and so the underreporting ratio is poorly estimated.

Minor Comments

Abstract mentions ‘extended capital region’ its not clear what this is ?

Abstract: infection statistics are not really necessary in the abstract . These can be removed.

Page 5 line 105 Notifications of what? Its not defined.

Page 5 line 109 How were the symptom onset delays estimated over time? This is not explained

Page 10 line 219 ‘…have been observed…’

Page 12 line 260-261 are sigma and sigma_1 the same or not?

Reviewer #4: The authors use Bayesian Inference and three different sources of data – COVID 19 cases, serological surveys, and external data on antibody development to estimate time-dependent underreporting of COVID-19 cases during the first wave of the COVID-19 epidemic in Finland.

The authors measure the underreporting of SARS-CoV-2 infections as the ratio of two seroprevalences – (i) based on observations from the serosurveys, and (ii) estimated using the reported COVID-19 incidence and data on antibody development. The paper is interesting, written in detail and with sound modeling and analysis.

Some minor comments:

Abstract line 15: change ‘external data’ to ‘external data on antibody development’

How is the value of the delay from symptom onset to diagnosis, C set at 3.5 days? The authors later mention that the result is very sensitive to this value. However, some explanation as to why 3.5 days was chosen is warranted.

Provide some details on how the prior for (\\sigma and other parameters) are chosen? This could be done in the Estimation Model section or the Sensitivity analysis section. It is mentioned in the section ‘Sensitivity analysis’ that the data is not very informative about some parameters making the selection of model priors more salient.

Line 328-332: This could probably go in the Appendix/Supplementary Information

6. PLOS authors have the option to publish the peer review history of their article (what does this mean?). If published, this will include your full peer review and any attached files.

Reviewer #1: No

Reviewer #2: No

Reviewer #3: No

Reviewer #4: No

---

## [Author Response · Author response to Decision Letter 0]

1 Jun 2023

Journal Requirements:

TN: Our comments are prefixed with “TN: “. The line references in our comments refer to the manuscript file (without tracked changes).

TN: We have used PLOS ONE’s style including references to figures and tables, and those related to supplementary files. We have added the corresponding author initials.

2. We note that your ethics statement indicates "The study protocol was submitted for ethical review to the ethical review board of the Hospital District of Helsinki and Uusimaa. Written informed consent was obtained from all participants of the serological surveys." To ensure that your submission complies with our policy on human subject research (https://journals.plos.org/plosone/s/human-subjects-research) please clarify in the methods section of the manuscripts whether the ethical review board of the Hospital District of Helsinki and Uusimaa approved this study. If applicable please provide approval numbers.

TN: We have clarified in the methods section as follows: ”The study protocol was approved by the ethical committee of the Hospital District of Helsinki and Uusimaa (HUS/1137/2020).”

“This study was funded by the Finnish Institute for Health and Welfare.”

TN: We have clarified the financial disclosure statement in the cover letter.

“We report no conflict of interests related to the current work. Finnish Institute for Health and Welfare (THL) conducts Public-Private Partnership with vaccine manufacturers and has received research funding from Sanofi Inc., Pfizer Inc., and GlaxoSmithKline Biologicals SA for studies not related to COVID-19. Nieminen, Melin, Palmu and Jokinen have been investigators in these studies, but they have received no personal remuneration.”

TN: We have included an updated version of the competing interests statement in the cover letter.

TN: We have included an updated data availability statement in the cover letter.

TN: The phrases “data not shown” were in reference to data which were not a core part of the research presented. We removed those phrases from the manuscript.

7. Your ethics statement should only appear in the Methods section of your manuscript. If your ethics statement is written in any section besides the Methods, please move it to the Methods section and delete it from any other section. Please ensure that your ethics statement is included in your manuscript, as the ethics statement entered into the online submission form will not be published alongside your manuscript.

TN: We have included an ethics statement within the manuscript at the end of the Methods section.

8. We note that Figures 1 and 2 in your submission contain map images which may be copyrighted. All PLOS content is published under the Creative Commons Attribution License (CC BY 4.0), which means that the manuscript, images, and Supporting Information files will be freely available online, and any third party is permitted to access, download, copy, distribute, and use these materials in any way, even commercially, with proper attribution. For these reasons, we cannot publish previously copyrighted maps or satellite images created using proprietary data, such as Google software (Google Maps, Street View, and Earth). For more information, see our copyright guidelines: http://journals.plos.org/plosone/s/licenses-and-copyright.

 a. You may seek permission from the original copyright holder of Figures 1 and 2 to publish the content specifically under the CC BY 4.0 license. 

TN: The map figures 1 and 2 were created by the authors, using open-source software (R program). The figures are not previously copyrighted.

Reviewers' comments:

Reviewer's Responses to Questions

Comments to the Author

1. Is the manuscript technically sound, and do the data support the conclusions?

Reviewer #1: Yes

Reviewer #2: No

Reviewer #3: Yes

Reviewer #4: Yes

2. Has the statistical analysis been performed appropriately and rigorously?

Reviewer #1: Yes

Reviewer #2: Yes

Reviewer #3: Yes

Reviewer #4: Yes

3. Have the authors made all data underlying the findings in their manuscript fully available?

Reviewer #1: Yes

Reviewer #2: Yes

Reviewer #3: Yes

Reviewer #4: Yes

4. Is the manuscript presented in an intelligible fashion and written in standard English?

Reviewer #1: Yes

Reviewer #2: Yes

Reviewer #3: Yes

Reviewer #4: Yes

5. Review Comments to the Author

Reviewer #1: To assess under-reporting, the authors compared SARS-CoV-2 infections from serological surveys to reported COVID-19 illnesses during the first pandemic wave in Finland. Their analyses are rigorous, clearly described, and results are comparable to others (e.g., JAMA 2021; 326:1400-09, Lancet Reg Health Am 2023; 18:100403). As severity increases with co-morbidities, for which age is a proxy, and reporting increases with severity, I imagine that under-reporting of childhood infections was disproportionate. Could the authors stratify by age?

TN: Thank you for the comments. The serological surveys targeted adults only and we have no data on children. We have now clarified in the discussion that “Our study was limited to those 18–69 years old. For ethical reasons, the elderly most vulnerable to severe COVID-19 were not invited to participate during the beginning of the epidemic as participation required a medical site visit and therefore could increase the risk of infection with SARS-CoV-2. Children were not invited due to difficulties in obtaining informed consent from minors.”. (lines 505-).

TN: We also note that it indeed is likely that underreporting was higher in younger age groups, as the detected covid-19 cases showed a decreasing trend in age: “It is therefore likely that the underreporting was both higher and decreased more in the younger age groups during the first epidemic wave.” (lines 511-). We also updated the last paragraph of the discussion to further note the age limitation in our study.

TN: Stratification by age is generally a good suggestion. However, the very low number of confirmed positive samples available (only 7 total) restricts adjusted or stratified analyses in our case, as we comment in the discussion: “Other serological studies have used regression analysis or post stratification to account for differences in the age and sex distributions between the survey participants and the underlying population … We decided not to use such analytical methods due to the very small number of confirmed positive samples.” (lines 517-).

Reviewer #2: Review of “Underreporting of SARS-CoV-2 infections during the first wave of the 2020 COVID-19 epidemic in Finland – Bayesian inference based on a series of serological surveys”

by Tuomo A. Nieminen, Kari Auranen, Sangita Kulathinal, Tommi Härkänen, Merit Melin, Arto A. Palmu, Jukka Jokinen

Summary

In this manuscript, a Bayesian approach is used to estimate the underreporting of SARS-CoV-2 infections during the first wave of COVID-19 in Finland, that is, from March to June 2020. The analysis is based on a series of serological surveys.

It is estimated by the authors that there were 1 to 5 infections for every detected case during the first wave. Reporting is estimated to have been much poorer during the first months of this period (before April) with 4 to 17 infections for every detected case.

General comments

To estimate the underreporting of a disease is particularly important when the number of asymptomatic cases is high, as it is known to be the case for the COVID-19 disease. To do so, to use a series of serological surveys, when available as it is the case here, is a proper choice and to perform the analyses based on a Bayesian inference method appears also well designed for this purpose.

However, although the paper is well written and the analysis well driven, I found the results of rather limited reach. Maybe this methodology was not applied yet to COVID-19 in Finland, but it is rather common in itself (see e.g. [1-2]), and it has been applied to many other countries by other authors since the beginning of the pandemic (e.g. [3-5]). From our point of view, methodologically speaking, the approach is not sufficiently new to deserve a publication in PLoS.

TN: We agree that utilizing serological data to assess underreporting is not novel, however our analysis does include methodological novelty in terms of incorporating data on the time lag to developing antibodies. Our research work is worth publishing; our study clearly shows that in Finland the spread of SARS-CoV-2 was very limited during the early phases of the pandemic. 

TN: Our analysis is retrospective, but the methodology we describe can also be applied in real time to accumulating data. In fact, we originally applied the method in real-time during the beginning of the 2020 COVID-19 epidemic in Finland, and the results from our analysis informed about the spread of SARS-CoV-2 in Finland during that time. We added the following sentence to the end of the Discussion: “The analysis we here describe can also be applied in real time, and our method informed about the spread, detection, and severity of SARS-CoV-2 infection in Finland during 2020.” (lines 596-)

From an epidemiological point of view, these results are nice and of some interest, but to make it really useful, one would expect to have these results put in perspective with other factors and/or with explanations on the behaviors specifically observed.

In particular, results appear rather different from what was observed in other countries in Europe but the present analysis does not help to understand these differences. For instance, the prevalence in Finland is significantly lower than most of the other European countries. Can the underestimation contribute to explain such a behavior? Or may it result from dynamical reasons (a model was recently obtained showing that, some epidemiological systems can have a very different time evolution in amplitude under strictly the same sanitary conditions [6]) or due to specific policies (as it has been the case in several Asian countries)?

TN: We agree that the comparison of the seroprevalence estimates to those from other countries is relevant, and the possible reasons are worth discussing. Finland had the advantage of being slightly isolated from mainland Europe and thus the epidemic started a few weeks later and never really developed into much of an epidemic during spring 2020, as implicated by our study. 

TN: The general public’s compliance with recommendations was likely very good due to Finland being a high-trust society. The political decisions during spring 2020 were also rather extreme, for example the whole extended capital area (Helsinki-Uusimaa region) was isolated from the rest of the country for several weeks. The result was that there was a large reduction in the daily numbers of social contacts in the early part of the 2020 COVID-19 epidemic in Finland, which was likely a major contributor to the steady decline of the epidemic in the country (Auranen et al. 2021). We have now added a paragraph to the end of discussion where we discuss the differences to other European countries as well as the possible reasons for these differences (lines 582-).

TN: Of note, another study utilising the same data shows that the prevalence of infection-induced antibodies remained at < 7% in Finland until the emergence of the Omicron variant at the end of 2021 (Solastie et al 2023).

Auranen, Kari & Shubin, Mikhail & Karhunen, Markku & Sivelä, Jonas & Leino, Tuija & Nurhonen, Markku. (2021). Social Distancing and SARS-CoV-2 Transmission Potential Early in the Epidemic in Finland. Epidemiology (Cambridge, Mass.). Publish Ahead of Print. 10.1097/EDE.0000000000001344. 

Anna Solastie, Tuomo Nieminen, Nina Ekström, Hanna Nohynek, Lasse Lehtonen, Arto A. Palmu, Merit Melin. Changes in SARS-CoV-2 seroprevalence and population immunity in Finland, 2020–2022. medRxiv 2023.02.17.23286042; doi: https://doi.org/10.1101/2023.02.17.23286042

Here, a retrospective analysis is performed by the authors based on a serological survey, but such a serological was not available at the very beginning of the epidemic. To cope with this difficulty, other authors have used more basic approaches based on case fatality ratio [7]. What would have been the effect of such a rough approach in comparison to the (more robust) Bayesian approach here used?

I think these types of questions will deepen the investigations and make the discussions and the work interesting to a wider audience.

Despite its technical interest and quality, at this stage, I don’t think the present work can help much to understand the behavior observed in Finland in comparison to other countries in Europe or in the world. For this reason, I cannot recommend it for publication in a PLoS journal.

TN: We believe that the results implicated by this study are quite interesting. Our study implies that the incidence of SARS-CoV-2 infection was very low in Finland compared to other countries. We also present ideas for methodological development in the evaluation of underreporting of infections. As noted above, we have now added additional discussion related to the differences in seroprevalence compared to other countries (lines 582-). 

TN: The serological surveys were started quite quickly in Finland and the first samples were collected during early April 2020, only a month after the epidemic had started off. The analysis which we present here was originally performed in real-time during 2020. We have now added the following sentence to the last paragraph of discussion: “The analysis we here describe can also be applied in real time, and our method informed about the spread, detection, and severity of SARS-CoV-2 infection in Finland during 2020.”

TN: As noted above, there is also additional evidence now that indeed the spread of SARS-CoV-2 remained quite limited in Finland until the emergence of the Omicron variant (Solastie 2023).

Anna Solastie, Tuomo Nieminen, Nina Ekström, Hanna Nohynek, Lasse Lehtonen, Arto A. Palmu, Merit Melin. Changes in SARS-CoV-2 seroprevalence and population immunity in Finland, 2020–2022. medRxiv 2023.02.17.23286042; doi: https://doi.org/10.1101/2023.02.17.23286042

References

[1] M. Dvorzak and H. Wagner, Sparse Bayesian modelling of underreported count data, Statistical Modelling, 2016, 16, 24-46.

[2] Turbé H, Bjelogrlic M, Robert A, Gaudet-Blavignac C, Goldman JP, Lovis C. Adaptive Time-Dependent Priors and Bayesian Inference to Evaluate SARS-CoV-2 Public Health Measures Validated on 31 Countries. Front Public Health, 2021, 8, 583401.

[3] Lope DJ, Demirhan H. 2022. Spatiotemporal Bayesian estimation of the number of under-reported COVID-19 cases in Victoria Australia. PeerJ, 10, e14184 http://doi.org/10.7717/peerj.14184

[4] Paixão B, Baroni L, Pedroso M, Salles R, Escobar L, de Sousa C, de Freitas Saldanha R, Soares J, Coutinho R, Porto F, Ogasawara E. Estimation of COVID-19 Under-Reporting in the Brazilian States Through SARI. New Gener Comput., 2021, 39(3-4), 623-645.

[5] Ricardo Cao & José E. Chacón (2022) Introduction to the special issue on Data Science for COVID-19, Journal of Nonparametric Statistics, 34(3), 555-569.

[6] Thenon N, Peyre M, Huc M, Touré A, Roger F, Mangiarotti S (2022) COVID-19 in Africa: Underreporting, demographic effect, chaotic dynamics, and mitigation strategy impact. PLoS Negl. Trop. Dis., 16(9), e0010735.

[7] Russell Timothy W , Hellewell Joel , Jarvis Christopher I , van Zandvoort Kevin , Abbott Sam , Ratnayake Ruwan , CMMID COVID-19 working group , Flasche Stefan, Eggo Rosalind M , Edmunds W John , Kucharski Adam J . Estimating the infection and case fatality ratio for coronavirus disease (COVID-19) using age-adjusted data from the outbreak on the Diamond Princess cruise ship, 2020. Euro Surveill. 2020, 25(12), 2000256. https://doi.org/10.2807/1560-7917.

Detailed comments

*p.2 “If the virus causes clinical disease, the rate of […]”: Maybe I would say instead “If the virus causes specific clinical disease, the rate of […]”?

TN: Thank you, we changed the phrase as suggested.

*p.3 line 26 “the numbers of COVID-19 cases”: To avoid any misunderstanding I’d say “the numbers of COVID-19 new cases”

TN: Thank you, we changed the phrase to “the numbers of new COVID-19 cases”.

*p.4 line 1: Ref. [2] has been published now, since more than one year now… the authors did not update their bibliography before submitting the manuscript. See https://doi.org/10.1016/j.ijid.2020.12.038

TN: Thank you, we have updated to reference.

*p.4 line 53-60: Indeed, it is important to have information from other countries for comparison.

TN: We agree. We have now added an additional chapter to the discussion where we compare the seroprevalence observed in our study to those in a few other European countries (lines 582-)

*p.4 line 57-58 “[…] estimated that there were 11 SARS-Cov-2 infections for every COVID-19 case.”: Maybe I’d say “detected cases” instead of just “case”.

TN: We clarified as “… detected COVID-19 case.”.

*p.4, line 73: Here also, the Ref. [6] was not updated. See 10.1172/JCI138759

TN: The publication referenced by 10.1172/JCI138759 is a different publication with a very similar title. The content is different: this newer peer-reviewed publication does not include the data which we reference in our manuscript. Therefore, we keep to the original reference.

*lines 72-85: it seems it is here that you explain what will be investigated in the present study. But it is not very clear, we have to deduce it by reading the two paragraphs. I think, you should state it more clearly, maybe with “in this paper” at line 72 “To better address the delays in antibody responses, in this paper, we utilise […] »

TN: Thank you, we clarified as suggested.

*line 81-83 “The novelty of our methodology is in accounting for the uncertainty in the time lag from disease symptoms to seroconversion when estimating the time-evolving underreporting of infections. Our analysis shows how the underreporting of SARS-CoV-2 infections evolved over time during the first epidemic wave in Finland.”: It appears to be the main contribution of this paper. I find it rather narrow for an international publication.

TN: The main contribution is understood correctly. However, also the results; that the cumulative incidence of SARS-CoV-2 was very low in Finland during spring 2020 compared to other European countries, are also interesting. We have now added comparisons to the seroprevalences in few other European countries (lines 582-)

* lines 98-100 “These data consist of COVID-19 cases notified as either a positive SARS-CoV-2 finding from a microbiological laboratory or a clinical diagnosis by a medical doctor.”: If the two sources of information are separated, the analysis could be performed on the two datasets to investigate the robustness of the analysis.

TN: Restricting the analysis to the laboratory confirmed COVID-19 cases is a possible sensitivity analysis. However, as we note in the manuscript, the proportion of cases notified as a clinical diagnosis was very low, under 5% “Approximately 95% of the COVID-19 cases during the first epidemic wave in Finland were based on a positive SARS-CoV-2 finding from a polymerase chain reaction (PCR) test” (lines 99-101). Therefore, excluding the cases based on clinical diagnoses could not significantly affect our main results. We respectfully suggest that there is no need for this sensitivity analysis.

Reviewer #3: General comments

This paper focusses on the underreporting of SARS-CoV-2 during the first wave of the 2020 epidemic in Finland. It focusses on Bayesian inference based on serological surveys. It uses different data sources to identify infection rates.

The paper is well written and clear mostly. The method is applied to data for age group 18 – 69 but doesn’t included older people. Older people ,possibly in care homes, could have a higher risk of spread or infection. There must be a greater explanation/justification for the exclusion.

TN: Thank you for pointing out the need for justification of the target age group. We have clarified in the discussion as follows: “For ethical reasons, the elderly most vulnerable to severe COVID-19 were not invited to participate during the beginning of the epidemic as participation required a medical site visit and therefore could increase the risk of infection with SARS-CoV-2. Children were not invited due to difficulties in obtaining informed consent from minors.” (lines 505-)

Page 4 what is the extended capital region? Its not defined anywhere.

TN: We have changed “extended capital region” -> “Helsinki-Uusimaa region” in all places.

The focus is the estimation of the underreporting ratio which is a function of seroprevalence divided by case count. The seroprevalence is assumed to have a random walk prior distribution in the logit of the prevalence, and the count of tests is used to estimate the posterior distribution of the seroprevalence; whereas the case count is used in a binary model and the underlying

Table 3 displays the results of estimation for the seroprevalence from surveys and case counts.

There is an issue about this table however. A credible intervals is shown for the ratio (underreporting) but the two data streams are modelled separately. How can an interval be constructed for the ratio when these models are separately run using MCMC. The samples cant be shared.

TN: Thank you for the comment. Our estimate of underreporting is based on two separate models for seroprevalence. We post-process samples attained from these two separate models to attain samples from the distribution of the underreporting ratio. There should not be any issue with this. 

TN: Of note is that the underreporting ratio is not a parameter, but a posterior quantity. This comment let us find that we had misleadingly labeled the underreporting ratio as a parameter on S2 Table, which we have now fixed.

An interpretation issue: according to Table 3 while the ratio stabilizes from June onwards (around 2.5) the credible interval crosses 1.0 and so the underreporting is poorly estimated. This is not mentioned but is a serious problem.

In general for a ratio to be estimated with a credible interval the ratio should have been computed within a joint model for seroprevalence and case counts.

TN: In our current modeling approach, values of underreporting ratio below one have non-zero probability, i.e. they are possible. We can still interpret their meaning as there being no underreporting. We have added to the discussion: “Our estimation approach allowed values of the underreporting ratio below one, which would correspond to there being more COVID-19 cases than SARS-CoV-2 infections. This could occur in theory, in case the diagnosis procedure for COVID-19 (i.e. PCR test) was unspecific and the virus testing was widespread. Nevertheless, we believe this to be unrealistic in our study, and we simply interpret values below one to represent absence of underreporting.” (lines 458-).

TN: It is true that one could build a joint model in which one could incorporate additional assumptions related to the underreporting ratio; for example that it can only take values greater than one. This more complex model could describe the phenomenon more accurately. However, the absence of additional assumptions and the lack of a more complex analytical approach do not necessarily mean, in our opinion, that the underreporting ratio is poorly estimated. We do, however, agree that values of the underreporting ratio below one are unrealistic in this case, as noted above.

TN: We could, of course, choose the prior distributions differently. It would be possible to construct a more informative prior distribution in the Estimation model, which would result in narrower credible intervals for underreporting. We comment more on this in the sensitivity analysis (lines 411-) and also note in the discussion that a more informative prior distribution is a possibility (lines 452-). 

TN: Our study describes an analysis performed already during 2020, utilising knowledge/information available at the time, and the results were used to inform about the spread of SARS-CoV-2 in Finland in real time. We now note this fact in the last paragraph of discussion. Since current understanding of the epidemic in Finland is partly based on the data which we are presenting, using that knowledge for constructing very informative prior distributions would not seem appropriate in this case.

TN: Our current approach is a step forward in the methodology of the analysis of underreporting of infections during the early phases of an epidemic, and in the utilisation of different sources of information in such analyses. Future work could focus on yet more complex modelling of the phenomenon.

Finally I note that various prior parameters are assumed for the distributions included in the models and these are given in Table S1 for the estimation model, and while Table S2 shows effects of varying some of these, it is noticeable that at later time the credible interval for the ratio crosses 1.0 for most entries and so the underreporting ratio is poorly estimated.

TN: We agree that the underreporting ratio estimates from later times during the study are unreliable. We comment on this in the discussion as follows: “There is great uncertainty about the estimated seroprevalence and the corresponding estimate of underreporting at the end of the study period, due to the small number of samples available in the serosurveys.”. As noted above, we have now also added a note to the discussion that the Estimation model prior distribution could be more informative with regards the dependency between seroprevalences on consecutive weeks (lines 452-).

TN: See our previous comments regarding the underreporting ratio being poorly estimated. 

Minor Comments

Abstract mentions ‘extended capital region’ its not clear what this is ?

TN: We have changed “extended capital region” -> “Helsinki-Uusimaa region” in all places.

Abstract: infection statistics are not really necessary in the abstract . These can be removed.

TN: We think that the implications of the study with regards to the incidence of SARS-CoV-2 infection in Finland during 2020 are important, as the incidence was likely quite different compared to other European countries. Based on other comments, we have added discussion about the differences to other European countries to the second to last paragraph of Discussion.

Page 5 line 105 Notifications of what? Its not defined.

TN: We changed “Notifications” to “Records”, which are described in the previous sentence. Hopefully this is now clearer.

Page 5 line 109 How were the symptom onset delays estimated over time? This is not explained

TN: Thank you, this was indeed a bit unclear. At the time of the original analysis during 2020 (which we are describing in our current study), we only had available an expert evaluation of these delays. We have clarified that “According to expert evaluation during early 2020, the delay from symptom onset to COVID-19 diagnosis was deemed to be on average 3.5 days in the Helsinki-Uusimaa region.” (lines 108-110)

TN: However, later on we could verify the accuracy of this expert evaluation based on internal infection tracking data available at our institute, which allows for estimation in how the delay from symptom onset to diagnosis evolved in the capital city (Helsinki) during 2020. We do not have direct access to these data, but we asked for summary statistics to verify the accuracy of the expert evaluation of the delay. We now note in the Discussion that based on these internal data the expert evaluation was likely reasonably accurate (lines 572-).

Page 10 line 219 ‘…have been observed…’

TN: Thank you, we corrected the typo.

Page 12 line 260-261 are sigma and sigma_1 the same or not?

TN: These are different. We added a clarification to the text that these are indeed different parameters (line 258). Hopefully this helps. 

Reviewer #4: The authors use Bayesian Inference and three different sources of data – COVID 19 cases, serological surveys, and external data on antibody development to estimate time-dependent underreporting of COVID-19 cases during the first wave of the COVID-19 epidemic in Finland.

The authors measure the underreporting of SARS-CoV-2 infections as the ratio of two seroprevalences – (i) based on observations from the serosurveys, and (ii) estimated using the reported COVID-19 incidence and data on antibody development. The paper is interesting, written in detail and with sound modeling and analysis.

TN: Thank you for the comments.

Some minor comments:

Abstract line 15: change ‘external data’ to ‘external data on antibody development’

TN: We made this change as suggested.

How is the value of the delay from symptom onset to diagnosis, C set at 3.5 days? The authors later mention that the result is very sensitive to this value. However, some explanation as to why 3.5 days was chosen is warranted.

TN: We have now clarified in the methods and discussion that the choice of delay C = 3.5 was based on expert evaluation and data available during early 2020 (lines 108-110). We also mention in the discussion that this was likely a reasonably accurate estimate based on internal infection tracking data from the capital city (Helsinki), and we note that small variations in this delay do not significantly affect our analysis, and our results are not sensitive to small changes in the choice of delay C (lines 572-). 

TN: The individual variation in this delay is not accounted for in our analysis. But, unless the individual variation around those approximately 4 days is very significant, the effect to the analysis cannot be significant. 

Provide some details on how the prior for (\\sigma and other parameters) are chosen? This could be done in the Estimation Model section or the Sensitivity analysis section. It is mentioned in the section ‘Sensitivity analysis’ that the data is not very informative about some parameters making the selection of model priors more salient.

TN: Thank you for the suggestion. We have added a paragraph to the end of the “Estimation model” section, which describes the hyperparameter choices (lines 279-). We also made small edits to the Sensitivity analysis section to further clarify the effects of some possible choices (lines 411-).

Line 328-332: This could probably go in the Appendix/Supplementary Information

TN: This is a fine suggestion, but respectively, we would like to keep references to the computational methods in the main text as we feel that the implementation is of some relevance as well.

6. PLOS authors have the option to publish the peer review history of their article (what does this mean?). If published, this will include your full peer review and any attached files.

Do you want your identity to be public for this peer review? For information about this choice, including consent withdrawal, please see our Privacy Policy.

Reviewer #1: No

Reviewer #2: No

Reviewer #3: No

Reviewer #4: No

TN: We used PACE to process our figure files.

---

## [Editor Report · Decision Letter 1]

6 Jun 2023

Underreporting of SARS-CoV-2 infections during the first wave of the 2020 COVID-19 epidemic in Finland - Bayesian inference based on a series of serological surveys

PONE-D-22-33699R1

Dear Dr. Nieminen,

We’re pleased to inform you that your manuscript has been judged scientifically suitable for publication and will be formally accepted for publication once it meets all outstanding technical requirements.

Kind regards,

Timothy J Wade, Ph.D

Academic Editor

PLOS ONE
---

## [Editor Report · Acceptance letter]

13 Jun 2023

PONE-D-22-33699R1 

Underreporting of SARS-CoV-2 infections during the first wave of the 2020 COVID-19 epidemic in Finland - Bayesian inference based on a series of serological surveys 

Dear Dr. Nieminen:

I'm pleased to inform you that your manuscript has been deemed suitable for publication in PLOS ONE. Congratulations! Your manuscript is now with our production department. 

Kind regards, 

on behalf of

Dr. Timothy J Wade 

Academic Editor

PLOS ONE